# Conserved signalling functions for Mps1, Mad1 and Mad2 in the *Cryptococcus neoformans* spindle checkpoint

**Koly Aktar**[1], **Thomas Davies**[1☯], **Ioanna Leontiou**[1☯], **Ivan Clark**[1], **Christos Spanos**[1], **Edward Wallace**[1], **Laura Tuck**[1], **A. Arockia Jeyaprakash**[1,2], **Kevin G. Hardwick**[1]*

**1** Institute of Cell Biology, School of Biological Sciences, University of Edinburgh, Edinburgh, United Kingdom, **2** Gene Center, Department of Biochemistry, Ludwig Maximilians Universitat, Munich, Germany

☯ These authors contributed equally to this work.
* Kevin.Hardwick@ed.ac.uk

**Data Availability Statement:** Mass spectrometry data is linked to the PRIDE database, EBI under the title "Conserved signalling functions for Mps1, Mad1 and Mad2 in the Cryptococcus neoformans

## Abstract

*Cryptococcus neoformans* is an opportunistic, human fungal pathogen which undergoes fascinating switches in cell cycle control and ploidy when it encounters stressful environments such as the human lung. Here we carry out a mechanistic analysis of the spindle checkpoint which regulates the metaphase to anaphase transition, focusing on Mps1 kinase and the downstream checkpoint components Mad1 and Mad2. We demonstrate that *Cryptococcus mad1Δ* or *mad2Δ* strains are unable to respond to microtubule perturbations, continuing to re-bud and divide, and die as a consequence. Fluorescent tagging of Chromosome 3, using a lacO array and mNeonGreen-lacI fusion protein, demonstrates that *mad* mutants are unable to maintain sister-chromatid cohesion in the absence of microtubule polymers. Thus, the classic checkpoint functions of the SAC are conserved in *Cryptococcus*. In interphase, GFP-Mad1 is enriched at the nuclear periphery, and it is recruited to unattached kinetochores in mitosis. Purification of GFP-Mad1 followed by mass spectrometric analysis of associated proteins show that it forms a complex with Mad2 and that it interacts with other checkpoint signalling components (Bub1) and effectors (Cdc20 and APC/C sub-units) in mitosis. We also demonstrate that overexpression of Mps1 kinase is sufficient to arrest *Cryptococcus* cells in mitosis, and show that this arrest is dependent on both Mad1 and Mad2. We find that a C-terminal fragment of Mad1 is an effective *in vitro* substrate for Mps1 kinase and map several Mad1 phosphorylation sites. Some sites are highly conserved within the C-terminal Mad1 structure and we demonstrate that mutation of threonine 667 (T667A) leads to loss of checkpoint signalling and abrogation of the *GAL-MPS1* arrest. Thus Mps1-dependent phosphorylation of C-terminal Mad1 residues is a critical step in *Cryptococcus* spindle checkpoint signalling. We conclude that CnMps1 protein kinase, Mad1 and Mad2 proteins have all conserved their important, spindle checkpoint signalling roles helping ensure high fidelity chromosome segregation.

spindle checkpoint", ProteomeXchange accession: PXD052327.

**Funding:** This work was supported by grants from the Leverhulme Trust (RPG-2018-379 to IL, KGH); the Darwin Trust of Edinburgh (KA); the Wellcome Trust (SRF 202811 to AAJ; Wellcome Centre for Cell Biology core grant 203149; Wellcome iCM programme 218470 to TD; Wellcome Sir Henry Dale Fellowship 208779 to EW; the Wellcome-University of Edinburgh Institutional Strategic Support Fund for IC); and the European Research Council (ERC Advanced Grant CHROMSEG; 101054950, AAJ). The funders had no role in study design, data collection and analysis, decision to publish, or preparation of this manuscript.

**Competing interests:** The authors have declared that no competing interests exist.

## Author summary

*Cryptococcus neoformans* is an environmental fungus that kills a large fraction of AIDS patients through meningitis. The World Health Organisation recently published a report on disease-causing fungi and ranked *C. neoformans* at the top, in the critical priority group of fungal pathogens. This position reflects the public health importance of this organism, the lack of effective drugs for fighting its infection, and the need to prioritise its research and development.

*Cryptococcus* undergoes fascinating changes during its life and infection cycle. Here we study a control process regulating normal cell divisions, monitoring the movement of DNA between mother and daughter cells. This surveillance system is known as the spindle checkpoint and we demonstrate that its components (Mps1, Mad1 and Mad2), and their mechanism of action, are well conserved. Knowing this, we can now target subtle changes in this pathway for future drug development.

## Introduction

*Cryptococcus neoformans* was recently designated a critical priority human fungal pathogen by the WHO [1], being responsible for the death of around 20% of AIDS patients. Recent estimates have 152,000 annual cases of *Cryptococcal* meningitis worldwide, mainly in Sub-Saharan Africa, leading to ~112,000 deaths in AIDS patients [2]. Few drugs are currently available and drug-resistant strains have emerged in the clinic, so new treatments are urgently required [3]. We are investigating mitotic control of chromosome segregation as a possible future drug target in *Cryptococcus neoformans* [4].

*Cryptococcus neoformans* is a model basidiomycete. These are very distant from the other major division of fungi, the ascomycetes which include the well characterised model organisms *Saccharomyces cerevisiae* and *Schizosaccharomyces pombe*. Important previous studies have described cell division in *Cryptococcus neoformans*. It has 14 chromosomes [5] and undergoes a partially open mitosis with the nuclear division taking place in the bud and one set of chromosomes moving back to the mother cell during anaphase [6].

During human infection *Cryptococcal* cells are stressed and undergo fascinating morphological transitions. They can form a small morphotype, known as seed cells, which appear to help *Cryptococcus* enter organs beyond the lung [7]. They also form polyploid titan cells in the lung [8]. These large cells (up to 100 micron diameter) are protective: the immune system struggles to clear them because of their size and their protective outer capsule [9]. Titan cells are polyploid and a recent study identified a specific cyclin, Cln1, as a key regulator of this transition [10]. Once formed, titan cells continue to divide by budding off small daughter cells, which have a high viability and are haploid [8]. It has been proposed that this division is likely to be somewhat error-prone, leading to aneuploid daughters. Such aneuploidy would increase genetic diversity in the fungal population and thus be relevant to the generation of drug-resistance in the clinic [4]. In support of this, aneuploidy (chromosome 1 disomy) has been found to confer fluconazole resistance to *Cryptococcus* [11,12]. In general, aneuploidy leads to reduced fitness [13] and cells have evolved a number of surveillance systems, known as cell cycle checkpoints, that keep chromosome mis-segregation to a minimum.

The mitotic spindle checkpoint is a key regulator of both mitotic and meiotic divisions, and has been studied in detail in several model systems and human cells [14,15]. This cell cycle checkpoint monitors interactions between kinetochores and spindle microtubules, and if any problems are apparent the checkpoint provides additional time for them to be resolved, by

delaying the metaphase to anaphase transition. Its molecular components were identified in budding yeast genetic screens [16–18], and the *MAD*, *BUB* and *MPS1* genes and their mode of action have been found to be extremely well conserved through eukaryotic evolution [19]. The spindle checkpoint has not yet been described in *Cryptococcus*, although both the *BUB1* and *MPS1* kinases were shown to be relevant to virulence in a genome-wide screen where 129 protein kinases were knocked out [20].

Here we have knocked out the *MAD1* and *MAD2* genes and compared their phenotypes to that of the *mps1Δ* strain. We demonstrate that all three components are essential for spindle checkpoint function: in response to anti-microtubule drug treatment, deletion mutants are unable to maintain sister-chromatid cohesion, continue to divide and die. Overexpression of CnMps1 kinase is sufficient to induce metaphase arrest and this is dependent on both Mad1 and Mad2. We identify Mps1 phosphorylation sites in the Mad1 C-terminus and demonstrate that the C-terminal Mad1 T667A substitution abolishes checkpoint signalling and *GAL-MPS1* arrest. We conclude that many aspects of spindle checkpoint signalling and Mps1, Mad1 and Mad2 functions are conserved in basidiomycetes.

## Results

Analysis of the *Cryptococcus neoformans* genome sequence in the FungiDB database identified CNAG_04824 and CNAG_01638 as likely homologues of *MAD1* and *MAD2*.

Their sequences are well conserved, with CnMad1 being predicted by alphafold to be an elongated coiled-coil protein, likely to bind both Mad2 and Bub1 (Fig 1A). To test whether their function is conserved we knocked out both genes, using the *amdS* Blaster (dominant recyclable marker) approach, developed in *Cryptococcus* by James Fraser et al [21]. First we replaced the *MAD* gene with the *amdS* marker by homologous recombination (S1A Fig). *AmdS*, encoding the *Aspergillus nidulans* acetamidase gene, enables acetamide to be used as both a carbon and nitrogen source by cells, and transformants were selected for on acetamide plates. In a second step the *amdS* marker was allowed to recombine out, via flanking repeat sequences, and its loss was selected for by growth on fluoracetamide. This compound kills any cells still containing the *amdS* marker as it is metabolised into toxic fluoroacetyl CoA and fluorocitrate, disabling aconitase and inhibiting the citric acid cycle. PCR analysis of genomic DNA was used to confirm targeted integration and recombination (S1B Fig). To confirm the *mad1* deletion, we made a polyclonal anti-Mad1 antibody, by expressing amino acids 1–200 of Mad1 fused to 6xHis-MBP (maltose binding protein) in bacteria and injecting the purified MBP-Mad1 protein as antigen into a sheep. The resulting serum was affinity-purified and then used to immunoblot Mad1 proteins from several *Cryptococcus* strains (Fig 1B) confirming that Mad1p is no longer expressed in the *mad1Δ* strain.

To analyse the loss of function phenotype we plated *mad1Δ* cells on the anti-microtubule drug benomyl. Fig 1C shows that the *mad1Δ* strain is sensitive to this drug, but not as sensitive as an *mps1Δ* strain. We used *mps1Δ* as a control: this strain has been previously described by the Bahn group, when analysing the kinome of *Cryptococcus neoformans* [20]. Fig 1C shows that we can rescue the benomyl sensitivity of the *mad1Δ* strain by complementation with a GFP-Mad1 construct, confirming that we have knocked out the right gene and that the benomyl sensitivity observed is due to loss of Mad1 function. Fig 1B shows that the expression level of this GFP-Mad1 fusion protein is similar to wild-type Mad1 levels and Fig 1C shows that the GFP-Mad1 fusion protein is functional.

Next we knocked out the *MAD2* gene using the same approach, and confirmed homologous recombination through PCR analysis of genomic DNA, before and after the *amdS* marker was lost (S2 Fig). These strains were also found to be benomyl sensitive and could be rescued

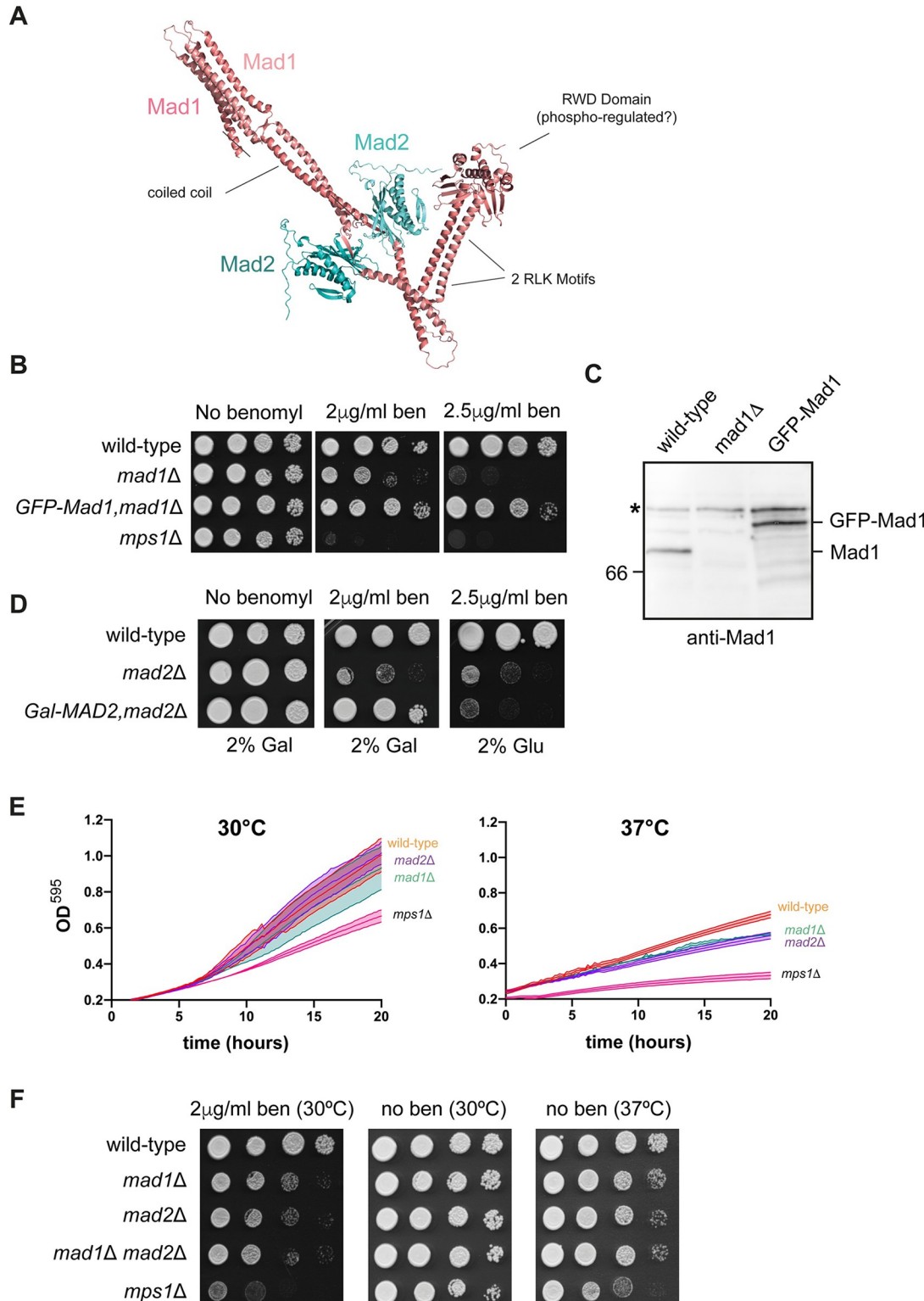

**Fig 1. CnMad1 and Mad2 proteins are conserved and *mad1* and *mad2* mutants are sensitive to anti-microtubule drugs.** (A) An alphafold model of Mad1-Mad2 hetero-tetramer indicating conserved domains in CnMad1: the Mad2 interaction site; 2 putative Bub1 binding motifs (RLK) and the structurally conserved C-terminal domain (RWD). (B) The *mad1*Δ strain is benomyl sensitive and can be complemented with an ectopic GFP-Mad1 construct. Strains were grown on YPD plates containing the indicated concentrations of benomyl for 3 days at 30˚C. (C) Anti-Mad1 immunoblot confirms the *mad1*Δ strain.

The three strains indicated were grown to log phase, harvested and whole cell extracts generated. The upper band (~115kD, labelled *) recognised by the anti-Mad1 antibody is a cross-reacting band and is used here as a loading control. (D) Benomyl sensitivity of the *mad2Δ* and complementation with ectopic *GALp-MAD2*. Strains were grown on plates (with 2% glucose or 2% galactose) containing the indicated concentrations of benomyl for 3 days at 30˚C. (E) Plate reader experiments confirm that *mad1*, *mad2* and *mps1* mutants are all temperature sensitive, with populations expanding slower than wild-type. Cells were diluted to $OD_{600}$ 0.2 and grown with shaking at the indicated temperatures. For viability assay (colony forming units) see S3 Fig. (F) The *mad1,mad2* double mutant does not display a synthetic phenotype. *mad1*, *mad2*, *mad1,mad2 double* and *mps1* strains were diluted, plated and grown on YPD plates at the indicated temperatures and drug concentrations for 3 days.

by expression of an ectopic copy of Myc-*MAD2* expressed from the *GAL7* promoter in galactose media (Fig 1D).

We next made a *mad1*, *mad2* double mutant, by repeating this experiment and sequentially knocking out *MAD2* in the *mad1Δ* strain. Fig 1F shows that the double mutant (*mad1*, *mad2*) was no more sensitive than the single mutants, suggesting that these proteins carry out their function(s) in a concerted fashion. This is consistent with known behaviour of these Mad proteins in other systems where they form a stable, constitutive complex [22–25].

As expected [26], the strain lacking Mps1 protein kinase is significantly more sensitive to the anti-microtubule drugs than the *mad* mutants (see Fig 1E). Our interpretation is that CnMps1 kinase is likely to have other mitotic functions, in addition to checkpoint signalling, such as error-correction and bi-orientation [27–29]. *mps1Δ* mutants were also reported to be temperature sensitive [20]. We found a subtle temperature-sensitive phenotype on plates for the *mad* mutants (Fig 1E), and Fig 1F shows that growth of liquid cultures in a plate reader confirms the phenotype at 37˚C for both *mad1* and *mad2* mutants. This *ts* phenotype was not as severe as that displayed by the *mps1Δ* mutant. Thus, as expected of a spindle checkpoint component, the *mad1* and *mad2* mutants are sensitive to anti-microtubule drugs. However, they are also sensitive to other stresses, such as high temperatures, which to our knowledge has not been reported in other yeasts.

## *mad1Δ* and *mad2Δ* mutants are checkpoint defective

To test whether these mutants have spindle checkpoint defects, we carried out additional assays that analyse how individual cells respond to anti-microtubule drug treatment. More specifically, we asked whether they can arrest in mitosis and maintain sister-chromatid cohesion when treated with the microtubule-depolymerising agent nocodazole. Fig 2A and 2B reveal that, unlike wild-type (H99) cells, *mad1Δ*, *mad2Δ*, or *mps1Δ* mutants were unable to arrest as large-budded cells when treated with nocodazole. Wild-type cells display a very robust arrest under these conditions (2μg/ml nocodazole treatment for 90 minutes at 30˚C). A subtle mitotic delay, in response to nocodazole treatment, might be missed with such fixed time point analysis, so we also employed microfluidics to analyse in more detail how single cells respond to nocodazole over time. Strains were pre-grown in SC media and then put in the microfluidics device, where single cells were captured (Fig 2C) before imaging. After 5 hours, nocodazole was added (2μg/ml) and imaging continued for a further 6 hours, with images being captured every 2 minutes. Movies were then analysed to see how many individual cells arrested in the nocodazole, and how many continued to re-bud and divide. Fig 2D confirms that the *mad1Δ*, *mad2Δ*, and *mps1Δ* mutants were all unable to delay mitotic progression in response to nocodazole treatment. We conclude that deletion of any one of these genes has likely completely abrogated the checkpoint response. Our results are in agreement with those from the Sanyal lab which independently generated a *mad2Δ* strain and demonstrated that it was unable to arrest as large-budded cells upon thiabendazole treatment [30].

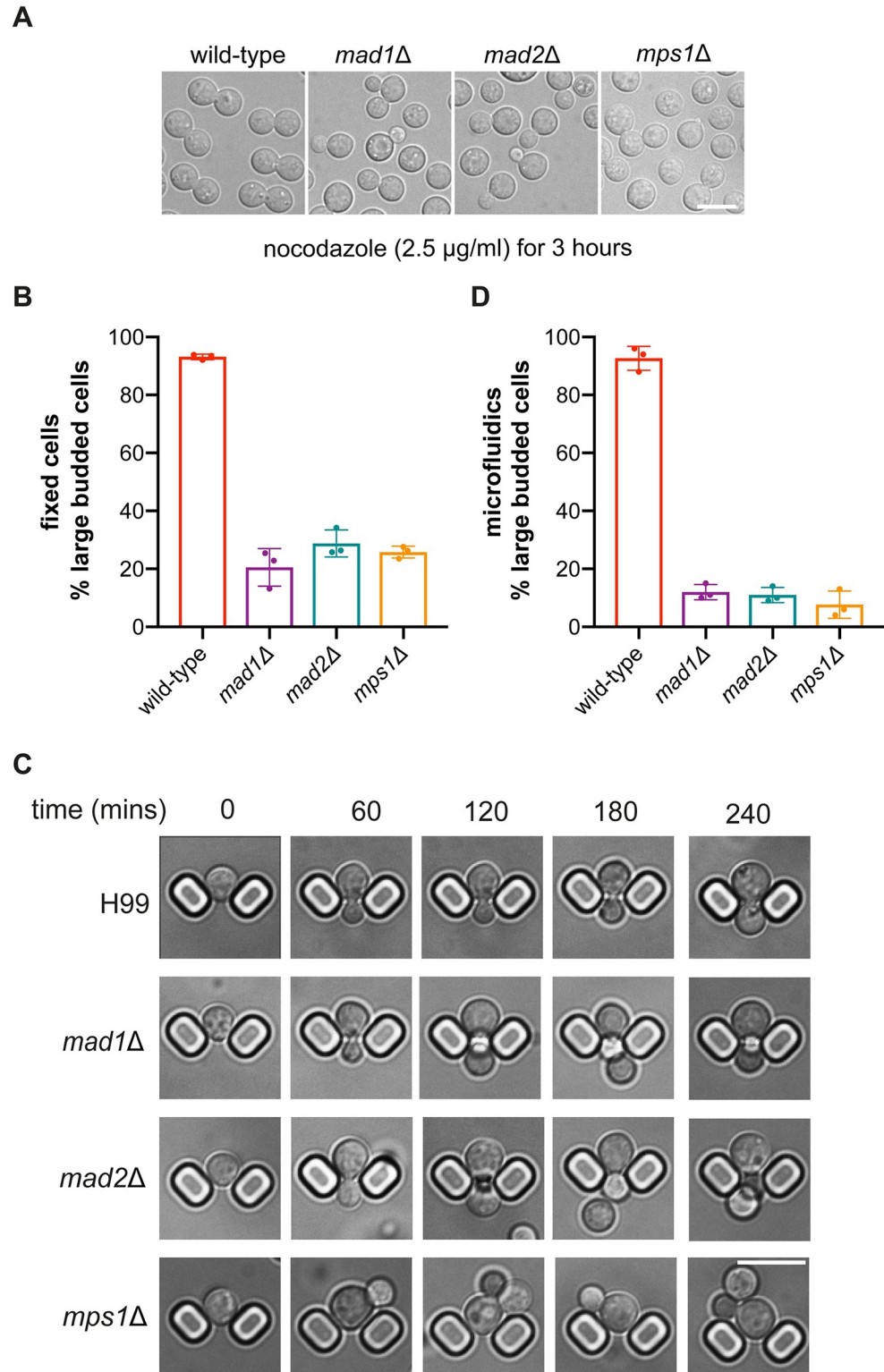

**Fig 2. Cn*mad1Δ* and *mad2Δ* strains fail to checkpoint arrest.** (A) *mad1*, *mad2* and *mps1* strains fail to maintain a large-budded nocodazole arrest. The strains indicated were grown to log phase and then nocodazole was added to a final concentration of 2.5 μg/ml in YPD media for 3 hours. Scale bar is 10μm. (B) Quantitation of the large-budded cells (bud diameter >4μm) observed at the 3 hour time point during this nocodazole challenge. This experiment was repeated 3 times and 300 cells were counted for each strain per experiment. (C) Images taken from a microfluidics

analysis of this behaviour in nocodazole (for H99, *mad1*, *mad2 and mps1* mutants). Cells were pre-grown in synthetic complete media supplemented with 0.2g/ml glucose and 0.05% w/v bovine serum albumin (Sigma) and then injected into the microfluidics device and analysed for 4 hours (+/- 2.5μg/ml nocodazole). Scale bar is 10μm. (D) Quantitation of the microfluidics experiment: movies were analysed manually for re-budding. This experiment was repeated three times and at least 100 cells counted per strain each time.

The key roles of the SAC are to delay the metaphase to anaphase transition and to protect sister-chromatid cohesion. If sister-chromatids separate prematurely, before bi-orientation of all chromosomes is complete, then sisters will segregate randomly leading to aneuploidy. GFP-marked chromosomes are widely used in model organisms to monitor and quantitate sister-chromatid separation [31,32]. We integrated an array of 240 lac operators (lacO, [33]) into chromosome 3 and expressed a lacI-NeonGreen fusion protein to mark just one of the fourteen *Cryptococcus* chromosomes (Fig 3A). We knocked out *MAD2* in this strain as above (producing KA159) and then complemented that strain with an ectopic copy of *MAD2* expressed from its own promoter (producing KA196). Comparison of these strains in the presence of nocodazole demonstrates a sister-chromatid cohesion defect in the *mad2Δ* strain (Fig 3B). After 90 minutes of nocodazole treatment ~40% of *mad2Δ* cells have separated sisters, compared to only 2% of wild-type cells in (Fig 3C). It should be noted that in such assays not all sisters will visibly separate, as in the absence of microtubules there are no spindle forces pulling them apart.

Next, we analysed the rate of death of the *mad* mutants in response to anti-microtubule drug treatment. To quantitate this, we determined the number of colony forming units (CFU) on plates following a time-course of nocodazole treatment for wild-type and *sac-* mutant cultures. The *mad* and *mps1* mutants all die faster than wild-type cells, likely as a consequence of their first division in nocodazole (S3 Fig).

In summary, in response to anti-microtubule drug treatments, the *mad* mutants fail to arrest, fail to maintain sister-chromatid cohesion, re-bud and die. We conclude that *mad1*, *mad2* and *mps1* mutants are all spindle checkpoint defective.

## Mad1 localises to the nuclear envelope and is recruited to unattached mitotic kinetochores

In many systems the Mad proteins interact with the nuclear periphery throughout interphase and only get recruited to unattached kinetochores during mitosis [24,34–37]. Several reasons for this have been suggested, including that by keeping the Mad proteins away from the Bub proteins, which decorate chromosomes, it may prevent premature checkpoint signalling. Very early in vertebrate mitosis, before mitotic kinetochores have matured enough to be active in checkpoint signalling, nuclear pores act as a site of Mitotic Checkpoint Complex (MCC) assembly and thereby prevent premature anaphase onset [36]. Transport related functions at the nuclear envelope have been proposed for the Mad1 protein in *S.cerevisiae* [38]. We analysed GFP-CnMad1 through the cell cycle and carried out double label experiments with γ-tubulin and kinetochore markers (CENP-A, mCherry-Cse4 and mCherry-Dad2). Fig 4A shows that when the checkpoint isn't active GFP-Mad1 is enriched at the nuclear periphery. Representative mitotic images demonstrate that GFP-Mad1 is recruited to foci close to and then between spindle poles (labelled with gamma-tubulin) during spindle elongation (Fig 4B). During a nocodazole-induced checkpoint arrest, although close to spindle poles, Mad1 does not co-localise with gamma-tubulin (Fig 4C). Rather, in such arrested cells, GFP-Mad1 co-localises with centromere/kinetochore markers such as Cse4 and Dad2 (Figs 4D and S4). GFP-Mad1 does not co-localise with centromeres in interphase cells (Fig 4E). In *C.neoformans*

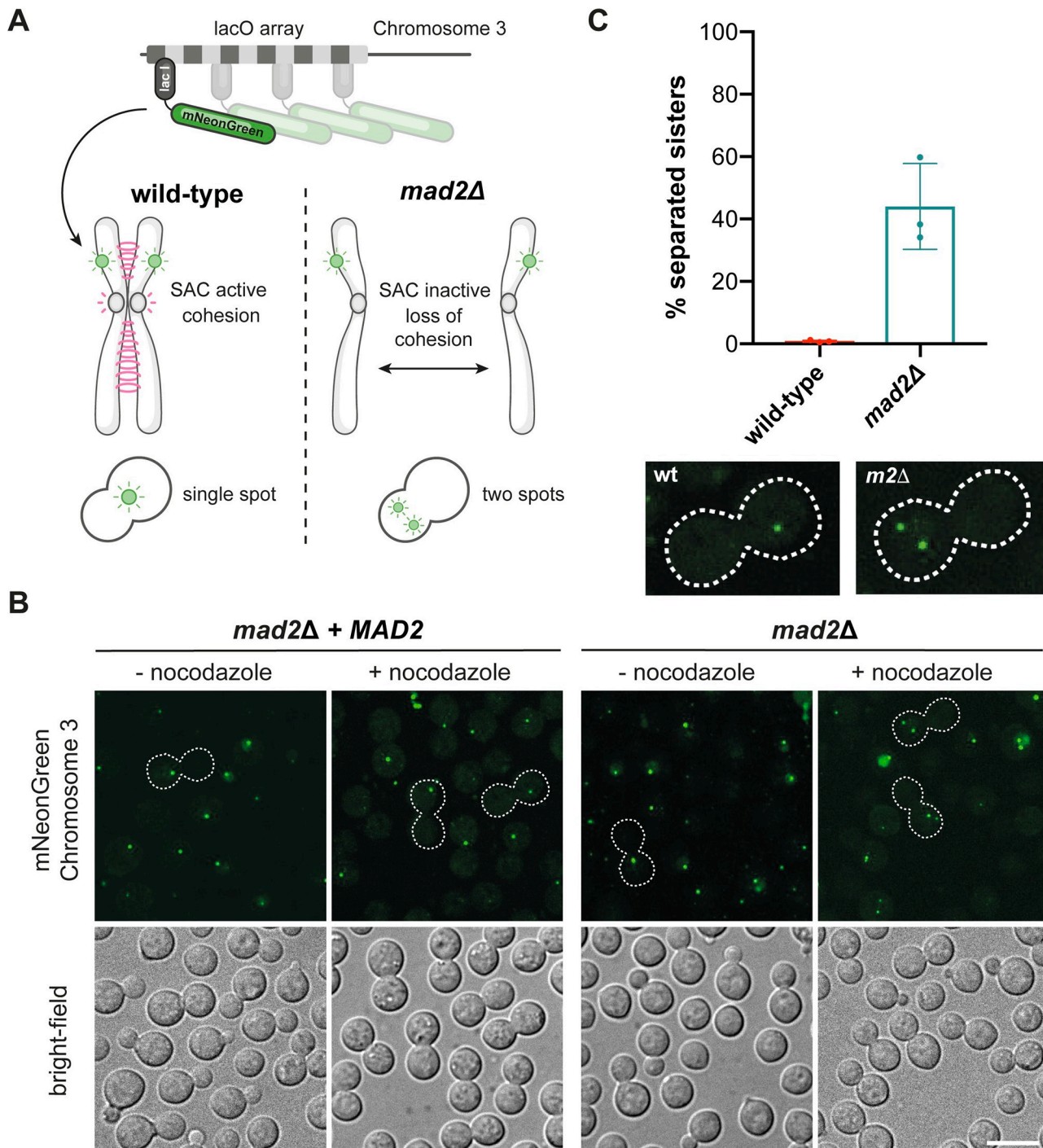

**Fig 3. The *mad2Δ* mutant fails to maintain sister-chromatid separation.** (A) Schematic representation of the strain with fluorescently-marked chromosome 3. An array of 240 lac operators were integrated at the safe-haven on chromosome 3 and lacI-mNeonGreen expressed. When sister chromatids separate 2 spots are seen. (B) mNeonGreen was imaged in YPD cultures of 'wild-type' (*mad2Δ*, complemented by an ectopic *MAD2* construct) and *mad2Δ* strains, 3 hours after the addition of 2.5μg/ml nocodazole. Scale bar is 10μm. (C) Quantitation of this experiment after three biological repeats. 100 cells were scored in each condition for each experiment. Note, as nocodazole is present there are no microtubule spindles to pull sister chromatids apart in this experiment, so not all sisters appear to separate. Images show relevant magnified cells from 3B.

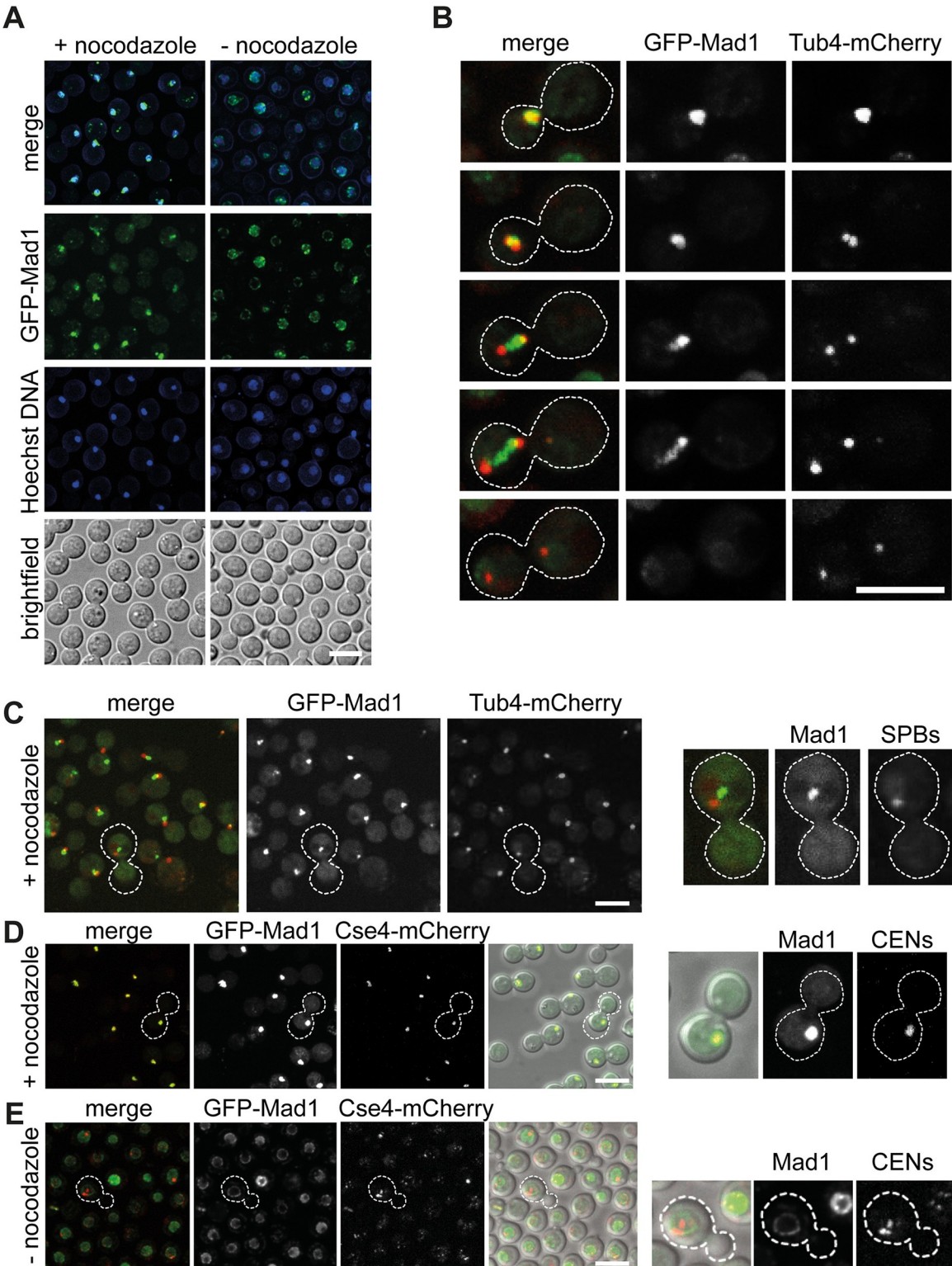

**Fig 4. CnMad1 accumulates at the nuclear periphery during interphase and is recruited to unattached kinetochores in mitosis.** (A) Strains expressing GFP-Mad1 were stained with Hoechst to label their DNA in cultures grown with and without the addition of 2.5μg/ml nocodazole. Scale bar is 10μm. (B) Five representative stages of mitosis are shown here, from live imaging of a strain expressing both GFP-Mad1 and mCherry-Tub4 (the gamma tubulin construct labels the spindle poles in mitosis). Scale bar is 10μm. (C) In nocodazole arrested cells, GFP-Mad1 is close to but does not co-localise with spindle poles (mCherry-Tub4). Scale bar is 10μm. (D) In nocodazole

arrested cells, GFP-Mad1 does co-localise with the centromere marker Cse4-mCherry. Scale bar is 10μm. (E) In interphase, cycling cells, GFP-Mad1 rarely co-localises with Cse4-mCherry. Scale is 10μm.

outer kinetochore proteins, such as mCherry-Dad2, only localise to kinetochores during mitosis [6], and in these cells co-localisation with GFP-Mad1 can be observed (S4 Fig). All of this is consistent with GFP-Mad1 being recruited to mitotic kinetochores in *Cryptococcus neoformans*, as it is in human cells and model ascomycete yeast systems [39].

## Mad1 interacts with several checkpoint components and effectors

To understand more about the molecular interactions of GFP-Mad1 in *Cryptococcus neoformans* we carried out large scale Mad1 purifications, from cycling and from checkpoint arrested cells, and employed mass-spectrometry to identify associated proteins. We grew 500 ml of yeast in YPD, added nocodazole for 3 hours, then harvested cells, made whole cell extracts and immunopurified GFP-Mad1 using GFP-TRAP beads. After careful washing and elution, Mad1 and its interacting partners were analysed by mass spectrometry. The volcano plot in Fig 5A compares GFP-Mad1 tagged with an untagged control strain, both cultures having been arrested in mitosis with nocodazole. Proteins to the right of the vertical line (at zero on the x-axis) are specifically enriched with GFP-Mad1, when compared to the untagged control. Proteins above the horizontal line were detected with significant confidence (a -log$_{10}$P-value of 1.3 corresponds to a p-value cutoff of 0.05). This dataset demonstrates that Mad1 specifically interacts with other SAC components (Mad2, Bub1, Bub3, Mps1) and their key effectors Cdc20 [40,41]. Fig 5B compares GFP-Mad1 purifications from cycling and mitotic extracts. Here, along with the checkpoint components and Cdc20, many APC/C sub-units (Apc1,2,3,4,5,6,8 [42]) and polo kinase (Plk1) are enriched in the mitotic pull-down. The interaction between Mad1 and the APC/C is very striking, and is not observed in *S.cerevisiae*, *S. pombe* or human cells. The reason is that *Cryptococcus neoformans* Bub1 carries out all the functions of both Bub1 and BubR1/Mad3 and therefore forms part of the MCC-APC/C effector complex. In cycling cells, Mad1 interacted with several nucleoporins including the TPR/Mlp homologue (S5 Fig).

Live cell imaging (Fig 5D) confirmed that GFP-Mad1 and mCherry-Bub1 co-localised at unattached kinetochores, in nocodazole-arrested cells. Analysis of the Mad1 protein sequence (Fig 5C) identified two RLK motifs that might drive this interaction with Bub1 [43–45]. We made point mutations in both RLK motifs of GFP-Mad1 and tested whether mutant alleles were still able to rescue *mad1Δ*. S6 Fig shows that *mad1-RLK/AAA* (mutating the conserved RLK residues 567–569) mutant was unable to rescue the benomyl sensitivity of *mad1Δ*, demonstrating that the Bub1 interaction is important for Mad1 checkpoint signalling functions in *Cryptococcus*. Mutation of nearby RLK residues (549–551) that are not conserved had no impact on function. We confirmed Mad1-Bub1 complex formation by co-immunoprecipitation. Fig 5E demonstrates that whilst wild-type GFP-Mad1 co-immunoprecipitates with mCherry-Bub1 in mitotic extracts, the GFP-*mad1-RLK/AAA(567–569)* mutant does not. We conclude that this key spindle checkpoint signalling interaction, between Bub1 and Mad1, is conserved in *Cryptococcus*.

**CnMps1 overexpression activates the checkpoint, arresting cells in mitosis in a Mad1 and Mad2-dependent fashion.** In several systems, overexpression of Mps1 kinase is sufficient to activate checkpoint signalling and arrest cells in early mitosis [46,47]. Sometimes it does this without significantly perturbing the mitotic machinery and cells simply divide at a slow rate due to a prolonged mitotic delay each cell cycle [46]. This has been a very useful tool in other systems, so we assembled a $P_{GAL7}$-Myc-MPS1 construct that does not express in YPD

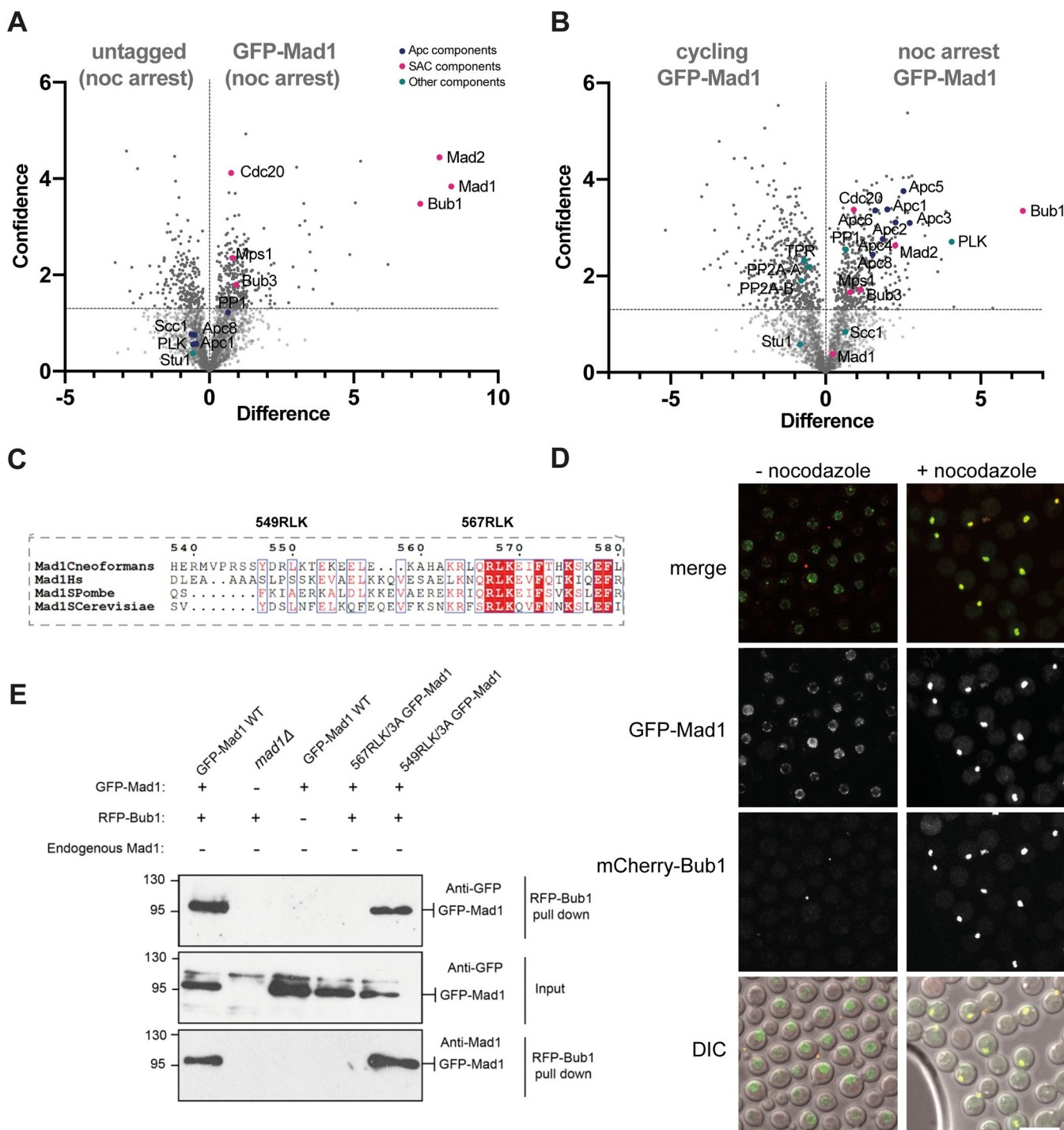

**Fig 5. CnMad1 interacts with Mad2, Bub1, Bub3 and several mitotic regulators including Cdc20-APC/C.** GFP-Mad1 was immunoprecipitated from extracts using GFP-TRAP and immunoprecipitates were run into an SDS-PAGE gel, cut out and digested into peptides with trypsin before analysis on a Orbitrap Fusion Lumos Tribrid mass spectrometer. (A) Analysis of GFP-Mad1 interactors: untagged, nocodazole-arrested cells v. nocodazole-arrested GFP-Mad1. This volcano plot shows both the difference (mean, label free quantitation LFQ difference, on the x-axis) and their statistical confidence (-log$_{10}$P-value of Perseus statistical test, on the y-axis) between polypeptides identified in the GFP-Mad1 and untagged control pull-downs (n = 3 for each purification). The -log$_{10}$P-value of 1.3 used here corresponds to a p-value cutoff of 0.05. (B) Analysis of mitotic GFP-Mad1 interactors: cycling GFP-Mad1 cells v. nocodazole-arrested GFP-Mad1. (C) Alignment of the Mad-RLK region in *C. neoformans*, *Homo sapiens*, *Schizosaccharomyces pombe* and *Saccharomyces cerevisiae*. Only the second RLK (residues 567–9) is within a conserved region. (D) GFP-Mad1 and mCherry-Bub1 co-localise in mitotic cells. Scale bar is

10μm. (E) Co-immunoprecipitation of Mad1 and Bub1 is Mad1-RLK-dependent. mCherry-Bub1 was immunoprecipitated from the five strains indicated. Bub1 immunoprecipitates were washed, split into two sets, separated by SDS-PAGE, then immunoblotted with anti-GFP and anti-Mad1 antibodies.

(glucose media) but expresses Mps1 at high levels in galactose media (Fig 6A–6C). This strain also expresses GFP-tubulin and analysis of cultures demonstrated a very robust metaphase arrest with 60% short mitotic spindles 3 hours after addition of 2% galactose, and >80% arrest after 5 hours. Carrying out the same experiment in *mad1Δ* or *mad2Δ* cells (Fig 6D and 6E) led to <10% metaphase cells, demonstrating that the $P_{GAL7}$-*MPS1* induced checkpoint arrest is dependent on both Mad1 and Mad2. For the *mad* mutants, no difference in metaphase counts were observed when comparing glucose with galactose media (not shown). For technical reasons the latter strains did not express GFP-tagged tubulin. Instead we scored metaphase cells as those with large buds, containing a single, DAPI-stained nucleus in the bud (as is typical for basidiomycetes [6]).

We were surprised to observe that all of these strains (wild-type, *mad1Δ* and *mad2Δ*) died within one or two divisions after galactose addition. Future experiments will be needed to understand what kills these cells. Intriguingly, we also found that overexpression of kinase-dead Mps1 was very toxic.

**The Mad1 C-terminus is phosphorylated by Mps1 and this is required for checkpoint arrest.** Mad1 is an important Mps1 substrate for SAC signalling [46,48]. To test whether Mad1 is phospho-regulated in *Cryptococcus*, we carried out *in vitro* Mps1 kinase assays. The CnMps1 kinase domain was expressed and purified as a 6xHis-Sumo-tagged fusion protein (residues 478–842) from bacteria. Radioactive assays employing this recombinant kinase show that the Mad1 C-terminus (MBP-Mad1-CT, residues 324–679) is a good Mps1 substrate (Fig 7A and 7B). Mass spectrometric analysis identified 8 Mad1 phospho-peptides and T667/668 and T660/661 as candidate phosphorylation sites (S7 Fig). Alphafold modelling suggests that CnMad1 T667 is likely equivalent to T716 in human Mad1, previously shown to be important for Cdc20 interaction and checkpoint signalling (Fig 7C [48]). We made alanine substitutions in T660, T661, T667 and T668 and found that the mutation T667A leads to a strong benomyl-sensitive phenotype, much like the *mad1* deletion (Fig 7D and 7E).

Kinase assays with the double TT667/668AA *mad1* mutant confirm that these threonines are significant *in vitro* phosphorylation sites (with the mutant exhibiting a 30% reduction in $^{32}$P incorporation, Fig 7B). The *mad1-T667A* point mutant was unable to arrest in nocodazole (Fig 7F) and when $P_{GAL7}$-*MPS1* was overexpressed in galactose (Fig 7G). We conclude that the C-terminal phosphorylation site T667 is an important Mps1 substrate in *C. neoformans*. Similar Mad1 phosphorylation (of HsMad1 T716) enhances the Mad1-Cdc20 interaction in humans [48], but our preliminary *in vitro* studies using recombinant proteins have as yet failed to demonstrate a similar phospho-dependent interaction between the N-terminus of Cdc20 and the C-terminus of Mad1 in *Cryptococcus*.

## Discussion

The spindle checkpoint has not previously been analysed in mechanistic detail in basidiomycetes. Here we have demonstrated that the checkpoint signalling roles of Mps1 kinase, Mad1 and Mad2 and their molecular interactions are very well conserved in *Cryptococcus neoformans*. In a second study, we demonstrated that Bub1 and Bub3 functions are also very well conserved [49]. One important difference in the spindle checkpoint machinery is that *Cryptococcus* lacks a BubR1/Mad3 gene as, unlike in humans, *S.cerevisiae* and *S.pombe*, the CnBub1 gene hasn't undergone duplication [50]. CnBub1 carries out the functions of both Bub1 (as a kinetochore-bound, checkpoint signalling scaffold) and Mad3/BubR1 (as part of the diffusible

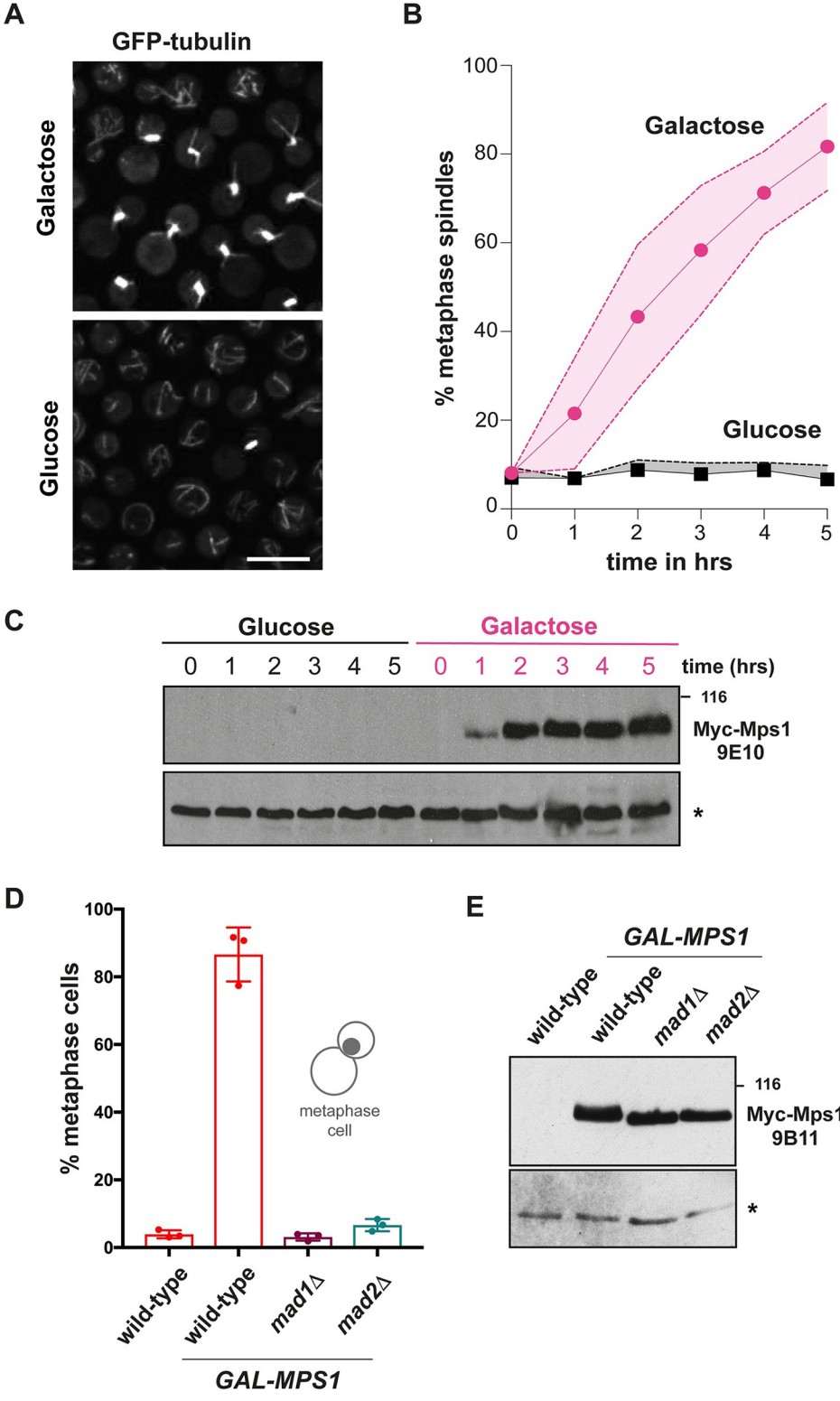

**Fig 6. Mps1 overexpression is sufficient to generate a mitotic checkpoint arrest.** (A) $P_{GAL7}$-*MPS1* was induced for 5 hours with 2% Galactose, or 2% glucose as the un-induced control. Scale bar is 10μm. (B) Quantitation of this mitotic arrest. Short spindles were scored every hour in at least 100 cells per condition at each time point. This experiment was repeated 3 times and displayed here as the mean +/- confidence levels. (C) Immunoblot of Mps1 levels at each hourly time point after Gal induction. Whole cell extracts were prepared by bead-beating, separated by SDS-PAGE and

immunoblotted with the 9E10 anti-myc monoclonal antibody. * indicates a cross-reacting band, used here as a loading control. (D) $P_{GAL7}$-*MPS1* metaphase arrest is Mad1- and Mad2-dependent. Quantitation of the % large-budded cells with a single, DAPI-stained nucleus in the bud at the 3 hour time point (experiment repeated 3 times and >100 cells counted per sample). (E) Immunoblot of Myc-Mps1 levels in wild-type and *mad* mutants. Note that the Mps1 protein has reduced gel-mobility in the wild-type extract as these cells are arrested in mitosis where Mps1p is heavily phosphorylated. The myc tag was detected here using the 9B11 monoclonal. * indicates a non-specific band used as a loading control.

APC/C inhibitor) [49]. Here we have shown that Mps1 phospho-regulation of the Mad1 C-terminus is also conserved in *Cryptococcus* [48]. Thus, much of what we have learnt about mitotic checkpoint signalling, from the past 30 years of studying model systems, is likely relevant in this important human fungal pathogen. We hope that this will enable us to efficiently identify specific checkpoint inhibitors and employ them in clinical treatments, in combination treatment with anti-microtubule drugs [51].

In interphase cells CnMad1 can be imaged at the nuclear periphery (Figs 4 and 5) and mass-spectrometric analysis of GFP-Mad1 pull downs identified several components of nuclear pores including the TPR/Mlp1 component of the inner basket (S5 Fig). Whilst this Mad1-TPR/Mlp interaction is conserved in many organisms, the role of Mad1-Mad2 localisation at the nuclear periphery remains unclear. This may keep Mad and Bub proteins apart in interphase to prevent inappropriate checkpoint signalling, and/or the Mad proteins could have a transport-related function at the nuclear pore [24,35,38,52]. Preliminary investigations suggest that the N-terminal coiled-coil (Fig 1) of Mad1 are important for this interaction with the nuclear envelope.

In mitotic cells GFP-Mad1 was recruited to unattached kinetochores, co-localising with Cse4, Dad2 and Bub1 centromere/kinetochore markers (Figs 4 and 5, and S4). Mass-spec identified Mad2, Bub1, Bub3, Mps1, Cdc20 and Apc1,2,3,4,5,6,8 as mitotic interactors highlighting Mad1's role at the centre of checkpoint signalling alongside Bub1 (Fig 5). In most organisms Mad1 is not found in a complex with the APC/C [10]. It does so in *Cryptococcus* through its interaction with Bub1. The *BUB1* gene did not duplicate in *Cryptococcus* and encodes a single polypeptide that carries out all the kinetochore-based signalling functions of Bub1, along with the MCC effector functions of BubR1/Mad3 including stable binding and inhibition of the APC/C [48].

We have demonstrated that the *mad* and *mps1* mutants are unable to arrest cells at metaphase in response to anti-microtubule drug treatment (Fig 2) and have developed a sister-chromatid cohesion assay which demonstrates that Cn*mad* mutants are unable to maintain cohesion of sister-chromatids during a nocodazole challenge (Fig 3). Thus these proteins all carry out critical checkpoint signalling function(s). Specific alleles of CnMad1 were generated, demonstrating that Bub1 binding (RLK mutant, S6 Fig) and Mps1-dependent phosphorylation (T667, Fig 7) are both important for checkpoint signalling and arrest. Thus Cryptococcus Mad1-Bub1 complex formation is likely central to forming a signalling platform for MCC assembly [45] and this might be enhanced by a phosphorylated Mad1 C-terminus, as in human cells [48]. As mentioned above, similar Mad1 phosphorylation (of human Mad1 T716) enhances the Mad1-Cdc20 interaction [48], However, our *in vitro* studies using recombinant proteins have as yet failed to demonstrate a similar phospho-dependent interaction between the N-terminus of Cdc20 and the C-terminus of Mad1 in *Cryptococcus*. There could be several reasons for this and we note that the CnCdc20 N-terminus is not particularly well conserved.

Overexpression of Mps1 kinase is a very useful tool for arresting cells in metaphase and dissecting downstream signalling (Fig 6). Here we confirm that both Mad1 and Mad2 act downstream of Mps1 kinase as both are needed for the overexpression of Mps1 to arrest

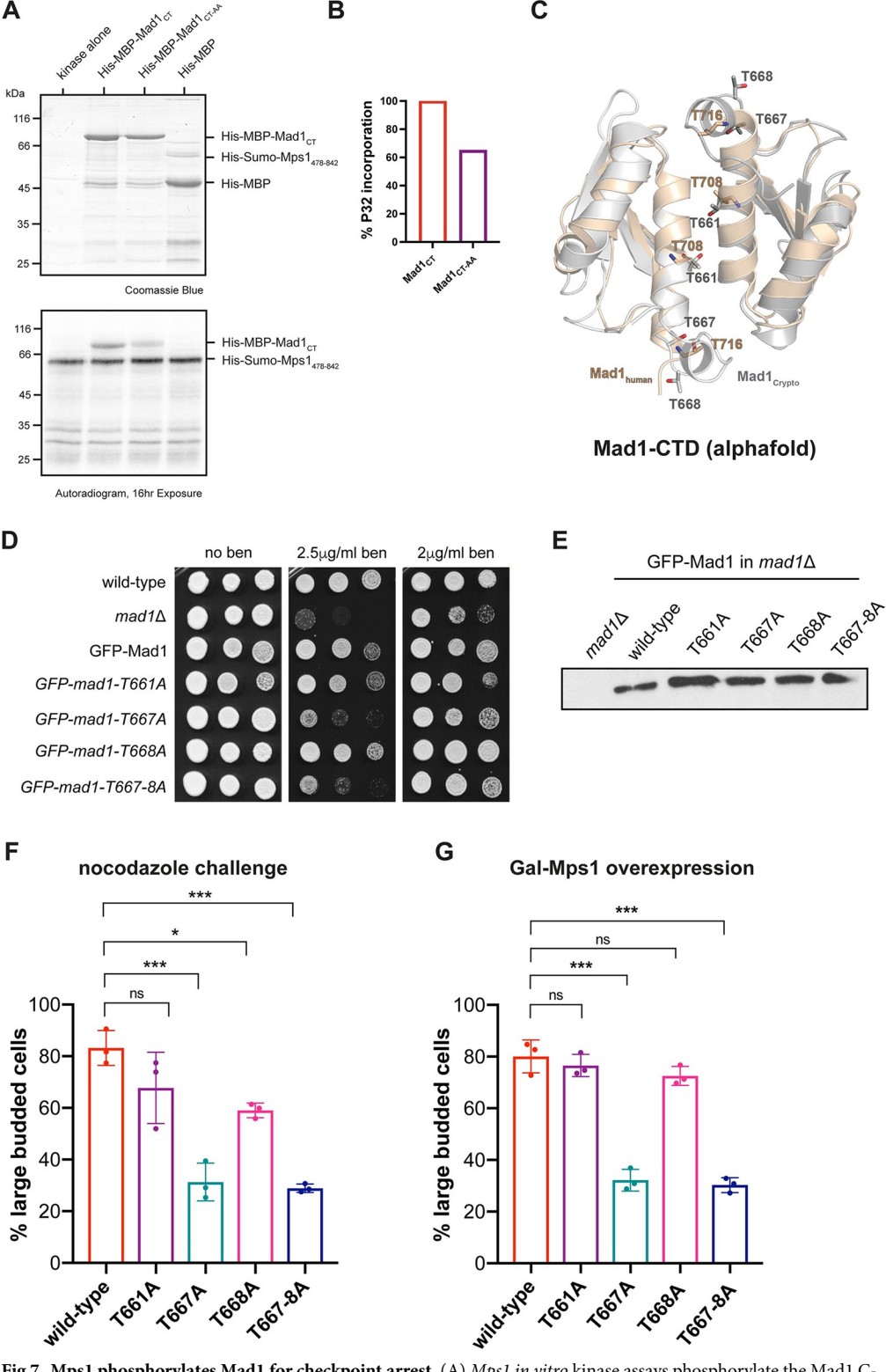

**Fig 7. Mps1 phosphorylates Mad1 for checkpoint arrest.** (A) *Mps1 in vitro* kinase assays phosphorylate the Mad1 C-terminus. His-MBP is used here as a negative (specificity) control. Mps1 autophosphorylates and phosphorylates the C-terminus of Mad1. See S7A Fig for phosphopeptides identified in CMad1. (B) Quantitation of the phosphorylation of the Mad1 C-terminus. The radioactive Mad1 signal here was normalised to the Mad1 Coomassie band. (C) Alphafold model of the CnMad1-CTD compared to the crystal structure of hsMad1-CTD. Relevant threonine residues are

highlighted on the model. (D) The *mad1-T667A* mutant, and the double (667,668) phospho-mutant, is benomyl sensitive. The strains indicated were grown at 30°C for 3 days. (E) Anti-GFP immunoblot of whole cell extracts. The GFP-*mad1* phospho-mutant proteins are stable. See S7B Fig for a related immunoblot using the anti-Mad1 antibody. (F) The *mad1-T667A* mutant, and the double phospho-mutant, fails to nocodazole arrest. Cultures were grown to log phase and then challenged with nocodazole for 3 hours. This experiment was repeated 3 times. (G) The *mad1-T667A* mutant, and the double phospho-mutant, fails to arrest when Mps1 kinase is over-expressed. Log phase cultures were induced with 2% galactose for 5 hours. This experiment was repeated 3 times.

cryptococcal cells. In *S.cerevisae*, wild-type cells can adapt to *GAL-MPS1* overexpression: they arrest every mitosis but after a few hours divide successfully to form viable colonies [46,53]. In preliminary experiments we have find that overexpression of CnMps1 is toxic, to both wild-type cells and to *mad* mutants. In addition, we find that overexpression of kinase-dead Mps1 is toxic. Similar results have been reported in *S.cerevisae* and this might help us to identify additional functions of CnMps1 kinase in future studies.

Our analysis indicates that the *mad1*, *mad2* and *mps1* deletion mutants all have a temperature-sensitive phenotype (Fig 1). To our knowledge, this has not been reported for *mad* mutants in other systems. Whilst several *mps1-ts* alleles were described in *S.cerevisae*, that is because Mps1 kinase has an essential function in spindle pole body duplication in that budding yeast [18]. However, a role for Mps1 in spindle pole duplication is not conserved in other systems [47,54]. Importantly, growth at 37°C is very relevant to human cryptococcal infections and so we are keen to see what impact these *mad/mps1* mutations have in infection models. Our preliminary data suggests that *mad* and *mps1* mutations do reduce the viability of *in vitro*-generated titan cells, and it has previously been demonstrated that mutation of Mps1 and Bub1 kinases reduces cryptococcal virulence during mouse infection [20]. Studies on the temperature-sensitive role(s) of Mad1, Mad2 and Mps1 during mitotic cell division and during titan cell formation are ongoing.

## Materials and Methods

### Yeast strains

*Cryptococcus* strains were all derived from *Cryptococcus neoformans* var. *grubii* H99 (Table 1).

### Yeast media

All *Cryptococcus* strains were generated in the H99 genetic background and stored in YPDA (2% bactopeptone, 1% yeast extract, 2% glucose, 2% agar, 1mM adenine) with 30% glycerol at -80°C. Strains were grown on YPDA plates for two days prior to use. For microfluidic experiments yeast synthetic complete media was used (SC: 0.67% Yeast Nitrogen Base; amino acid mix; 2% glucose). Knockout transformants generated using the blaster cassette were selected on plates containing YNB medium (0.45% yeast nitrogen base w/o amino acids and ammonium sulphate, 2% glucose, 10mM ammonium sulphate, 2% agar, 1mM adenine) with 5mM acetamide. To enable the blaster cassette to recombine out strains were streaked onto YPD twice and then YNB media containing 2% glucose, 10mM ammonium sulphate, and 10mM fluoroacetamide, as described previously[21]. For drug selection cells were grown on YPDA plates containing 300μg/ml hygromycin B (Invitrogen), 100μg/ml Geneticin (G418 Sulphate, Gibco) or 100μg/ml nourseothricin (Jena Bioscience). For expression from the *GAL7* promoter: it was switched on and off by adding 2% galactose or 2% glucose respectively.

Plate reader experiment (variable temperatures): log phase cells were diluted to OD 0.2 and grown for 20 hours in a Tecan i-control plate reader (infinite 200Pro), with time points taken every 10 minutes.

**Table 1. For the full list of strains generated in this study.**

| Name | Genotype |
|---|---|
| YSB3632 | *mps1Δ::NAT* [20]. |
| CNSD159 | *TUB4::TUB4-mCherry:G418* [30]. |
| CNSD105 | *DAD2::DAD2-mCherry:G418* [30]. |
| KA43 | *mad1Δ::amds* |
| KA51 | *mad1Δ* (no selection, as *amds* recombined out) |
| KA53 | *mad2Δ::amds* |
| KA55 | *mad2Δ* (no selection, as *amds* recombined out) |
| KA28 | *HISp:GFP-MAD1:HYG* (chrom 3, safe haven 3, pPEE37). |
| KA44 | *mad1Δ, HISp:GFP-MAD1:HYG* (chrom 3, safe haven 3, pPEE37). |
| KA77 | *mad1Δ, mad2Δ* |
| KA113 | *mad2Δ, GALp-myc-MAD2:NAT* (chrom 14, safe haven 7, pPEE31) |
| IL08 | *GALp-myc-MPS1:G418* (chrom 14, safe haven 7, pPEE36). |
| KA61 | *mad1Δ, GALp-myc-MPS1:G418* (chrom 14, safe haven 7, pPEE36). |
| KA63 | *mad2Δ, GALp-myc-MPS1:G418* (chrom 14, safe haven 7, pPEE36). |
| KA159 | *HISp-mNeonGreen-lacI:G418* (chrom 14, safe haven 7, pPEE36), *lacO-array:HYG* (chrom 3, safe haven 3, pPEE37), *mad2Δ:amds*. |
| KA196 | *HISp-mNeonGreen-lacI:G418* (chrom 14, safe haven 7, pPEE36), *lacO-array:HYG* (chrom 3, safe haven 3, pPEE37), *mad2Δ:amds, MAD2:NAT* (chrom 7, safe haven 5 pPEE29) |
| KA118 | *DAD2::DAD2-mCherry:G418, mad1Δ, HISp:GFP-MAD1:HYG* (chrom 3, safe haven 3, pPEE37). |
| KA139 | *TUB4::TUB4-mCherry:G418, mad1Δ, HISp:GFP-MAD1:HYG* (chrom 3, safe haven 3, pPEE37). |
| KA203 | *Cse4::CSE4-mCherry:G418, mad1Δ, HISp:GFP-MAD1:HYG* (chrom 3, safe haven 3, pPEE37). |
| KA208 | *HISp-mCherry-BUB1:NAT* (chrom 14, safe haven 7, pPEE31), *mad1Δ, HISp:GFP-MAD1:HYG* (chrom 3, safe haven 3, pPEE37). |
| KA163 | *mad1Δ, HISp:GFP-Mad1-T667A:HYG* (chrom 3, safe haven 3, pPEE37). |
| KA172 | *mad1Δ, HISp:GFP-Mad1-T668A:HYG* (chrom 3, safe haven 3, pPEE37). |
| KA177 | *mad1Δ, HISp:GFP-Mad1-T661A:HYG* (chrom 3, safe haven 3, pPEE37). |

## Gibson assembly, sequencing

Primers were purchased from either Sigma-Aldrich/Merck or IDT (all primers are listed in Table 2). Gibson assembly and NEBuilder HiFi DNA assembly (New England Biolabs) were performed following manufacturer's instructions; sequencing was carried out using Big Dye.

## *Cryptococcus neoformans* transformation protocol

Strains were pre-cultured in YPDA, then diluted overnight into 100ml of culture. These cultures were grown at 30°C (25°C for *bub1Δ*) until the $OD_{600}$ reached 0.3–0.36 the next morning. Cells were harvested, washed twice with 50ml iced cold water and once with 50ml of ice-cold electroporation buffer [10mM Tris-HCl pH7.5, 1mM $MgCl_2$ and 270mM sucrose]. Then cells were resuspended in 35ml electroporation buffer containing 150μl of 1M DTT and incubated for 15 mins on ice. Cells were then spun, washed with 50ml electroporation buffer, and finally resuspended in ~100μl of electroporation buffer. 40μl of cell suspension was gently mixed with linearised DNA (~4μg) and transferred to pre-cooled electroporation cuvettes (2mm). Electroporation was performed at 1400V, 600ohm and 25μF (Biorad, GenePulser). Following electro-pulse cells were incubated on ice for two minutes then 1 ml pre-warmed (30°C) YPDA was added to the cells before plating on YPDA. After overnight recovery, cells were replica-plated to selection plates. Colonies were typically tested by PCR for correct integration and then by western blot for protein expression.

**Table 2. Primers used in this study.**

| Primer number | Primer name | Sequence | Purpose |
|---|---|---|---|
| KA05 | KH_CNY | 5'_ATGTATGCAAGATGTATGCG_3' | Safe haven 3 |
| KA07 | HYG FORW 3 | CACTCGTCCGAGGGCAAAGG | Safe haven 3 screen |
| KA95 | M1D F1F | CTATAGGGCGAATTGGAGCTATTTGATCCAAGACGGGATC | *mad1Δ* |
| KA96 | M1D F1R | AATATAGTGGCATGATTGAAGAAAGAGGATATGGAGTTGC | *mad1Δ* |
| KA97 | M1D F3F | GATGGCTAGAGTAGAACTTATACAATCCAAATGTATATGTCG | *mad1Δ* |
| KA98 | M1D F3R | CTTGATATCGAATTCCTGCAACACGAAATTGAGCTCAC | *mad1Δ* |
| KA87 | M2D5F1bfT F | CCACCGCGGTGGCGGCCGCTATCCAGCTCGATCCATCTTG | *mad2Δ* |
| KA88 | M2D5F1bfTR | AATATAGTGGCATGATTGAAAGAATAAACATCATGTCTGCC | *mad2Δ* |
| KA89 | M2D3F3afT F | GATGGCTAGAGTAGAACTTAACTTCTTCTTTAACCGCTTG | *mad2Δ* |
| KA90 | M2D3F3afT R | AGGGAACAAAAGCTGGGTACGGTGGATGGACAAAATGAAG | *mad2Δ* |
| KA111 | M1 5 1100 | TGGGACGTACGATACGAGCGTTGAGAATTG | *mad1Δ* screen |
| KA112 | M1 3 1100 | TTTCACAGGTAACGCTCATCCCTGCAAAAA | *mad1Δ* screen |
| KA153 | Cse4mCF | TGGAGCTCCACCGCGGTGGCTCATGGAGAAGATAGATTGTATAG | Cse4 tagging |
| KA154 | Cse4mCR | GGATCCACTAGTTCTAGAGCGCTCAAATCGTAATCCTTC | Cse4 tagging |
| KA155 | M1CT F | TACTTCCAATCCAATGCAGATGCCGTAGGCGAAATGAGC | LIC Cloning |
| KA157 | M1CTiiR | TTATCCACTTCCAATGTTATTATCATCCAAGCCCGACATACCCAG | LIC Cloning |
| KA160 | HG F | TGGAGCTCCACCGCGGTGGCGGTACCGAGCTCGGCAGATAC | GFP-Mad1 tagging |
| KA161 | HG R | TTATTGGCATCATCTCTTCCGTGTTAATACAGATAAACCAAG | GFP-Mad1 tagging |
| KA162 | HGM1 F | GGAAGAGATGATGCCAATAACCCCGGCTC | GFP-Mad1 tagging |
| KA163 | HGM1 R | GGATCCACTAGTTCTAGAGCGCAAGTCAGGATAGCAGAGTG | GFP-Mad1 tagging |
| KA182 | TEF Promo F | TACCGTATAGCATAGAATGGC | Blaster screen |
| KA183 | Ade6 Term R | ATCATCTACTATATGCTTCGTAAATGTCCA | Blaster screen |
| KA186 | MIM F | GTGCAGGATTGTTACGAAGTTCGGCGCATGCTTCGTAAGCGGGGTTGAACTCT | Mutagenesis |
| KA187 | MIM R | AGAGTTCAACCCCGCTTACGAAGCATGCGCCGAACTTCGTAACAATCCTGCAC | Mutagenesis |
| KA188 | Mad2_5_1100kb F | ACACCGGCAAAGTCACTTTCAGATCCG | *mad2Δ* screen |
| KA189 | Mad2_3_1100kb R | CTTTCAGTGATCCTGAAGGAATCGAGCA | *mad2Δ* screen |
| KA256 | T661A_F | CAAACATCTCCGCTGTCACTTGTGCCAAAAACCCT | Mutagenesis |
| KA257 | T661A_R | AGGGTTTTTGGCACAAGTGACAGCGGAGATGTTTG | Mutagenesis |
| KA262 | T67A_F | AAAAAAAAAGAACCTACTGTAGCCTTCTCAAACATCTCCGTTGTC | Mutagenesis |
| KA263 | T67A_R | GACAACGGAGATGTTTGAGAAGGCTACAGTAGGTTCTTTTTTTTT | Mutagenesis |
| KA264 | T68A_F | GGAAAAAAAAAGAACCTACTGCAGTCTTCTCAAACATCTCCGT | Mutagenesis |
| KA265 | T68A_R | ACGGAGATGTTTGAGAAGACTGCAGTAGGTTCTTTTTTTTTTCC | Mutagenesis |
| KA320 | T667E For | CAGGGAAAAAAAAAGAACCTACTGTTTCCTTCTCAAACATCTCCGTTGTCACT | Mutagenesis |
| KA321 | T667E Rev | AGTGACAACGGAGATGTTTGAGAAGGAAACAGTAGGTTCTTTTTTTTTTCCCTG | Mutagenesis |
| KA322 | T668E For | AAAAAAAAAGAACCTACTTCAGTCTTCTCAAACATCTCCGTTGTCACTTGT | Mutagenesis |
| KA323 | T668E Rev | ACAAGTGACAACGGAGATGTTTGAGAAGACTGAAGTAGGTTCTTTTTTTTTT | Mutagenesis |
| IL222 | Gal7_Fwd | CGGCCGCTCTAGAACTAGTGAAGATATATATAGTAATAAATTTGAAATGAAC | Gal-myc-Mps1 Expression |
| IL226 | Gal7_rev_myc | ATTCAGATCCTCTTCAGAGATGAGCTTTTGCTCCATTCTCAGGAGAGAATTGAG | Gal-myc-Mps1 Expression |
| IL227 | Mps1_Fw_myc | ATGGAGCAAAAGCTCATCTCTGAAGAGGATCTGAATATGAGCTCACCCGAGAAC | Gal-myc-Mps1 Expression |
| IL231 | Mps1_rev | CGAATTCCTGCAGCCCGGGGTCCTGACATATGGACCTTG | Gal-myc-Mps1 Expression |
| TD32 | Mps1_481_LIC_Fw | TACTTCCAATCCAATGCAACTTTATTTCATGTGAACGGA | Kinase domain |
| TD34 | Mps1K_845_LIC_Rv | TTATCCACTTCCAATGTTATTATCTAGACAATGCGTTTTGAGC | Kinase domain |

## Knockout constructs

Both *mad1* and *mad2* knockout constructs were made using the Blaster construct [21]. 1 kb homologous arms, consisting of the *MAD* 5' and 3' UTR sequences, were ligated at either end of the selective amds2 marker: BlueScript vector was digested with HindIII/EcoRI, and pPEE8

**Table 3. Plasmids generated in this study.**

| KACP1 | mad1 deletion: Bluescript-MAD1-5'UTR(1kb)_amds2_MAD1-3'UTR(1kb) |
|---|---|
| KACP2 | mad2 deletion: Bluescript-MAD2-5'UTR(1kb)_amds2_MAD2-3'UTR(1kb) |
| KACP3 | pPEE37: HISp-GFP-*MAD1*:HygR |
| KACP4 | pPEE31: 5'UTR(1kb) mCherry-*CSE4*:NatR |
| KACP5 | pPEE37: 240xlacO-array:HygR |
| KACP6 | pPEE36: GPDIp_lacI-mNeonGreen:G418 |
| KACP7 | pPEE36: Gal7p_myc_*MPS1*:G418 |
| KACP8 | pPEE37: Gal7p_myc_*MAD2*:HygR |
| KACP9 | pPEE37: HISp_GFP_*mad1_549RLK/AAA*:HygR |
| KACP10 | pPEE37: HISp_GFP_*mad1-567RLK/AAA*:HygR |
| KACP11 | pPEE37: HISp_GFP_*mad1T661A*:HygR |
| KACP12 | pPEE37: HISp_GFP_*mad1T667A*:HygR |
| KACP13 | pPEE37: HISp_GFP_*mad1T668A*:HygR |

with SpeI/HindIII. The final construct for *mad1Δ* was digested with KpnI and AatII. For *mad2Δ*, SapI and SacII were used to digest the construct before transformation into the wild type H99 strain (see Table 3 for the full list of plasmid constructs). Transformed colonies were initially selected on acetamide plates. Cell were re-streaked several times to ensure stable integration and then streaked to single colonies to ensure that all cells contained the Blaster construct. Correct integration of the marker for *mad1Δ* was confirmed by PCR (S1 Fig) and by immunoblotting using the *Cryptococcus* anti-Mad1 antibody (Fig 1C). In a similar way, to confirm the *mad2Δ*, PCR analysis was performed with genomic DNA made from stable transformants (Fig 2B).

## Mad1 antibody generation and affinity purification

Residues 1–200 of Mad1 were amplified from cDNA and cloned into a LIC Biobrick vector (14C, 6xHis-MBP, https,//www.addgene.org/48284/). This Mad1 construct was expressed in *E. coli* pLysS cells (Agilent), purified on Talon Cobalt resin, eluted and then dialysed into 50mM Hepes pH7.6, 75mM KCl. The recombinant protein was used to immunise sheep (MRC PPU Reagents and Services, University of Dundee). Specific antibodies were affinity-purified using Affigel 10 resin coupled to the 6xHis-MBP-Mad1 protein. Sheep sera was diluted with PBS, filtered, then gently pumped through the Affigel-Mad1 column overnight. The column was thoroughly washed with PBS-Tween and then with PBS containing 0.5M NaCl. Finally, the specific anti-Mad1 antibodies were eluted at high pH, with 100 mM triethylamine (pH 11.5), and quickly neutralised before dialysing the antibodies overnight into PBS containing 40% glycerol.

## HISp-GFP-MAD1

For Gibson assembly of HISp-GFP-MAD1 the pCN19 vector was used to amplify GFP containing an intron. The HIS promoter and full length *MAD1* clone were PCR-amplified from H99 genomic DNA. The resulting three amplified fragments were Gibson assembled into the pPEE37 safe haven vector (developed by James Fraser). Plasmids were sequenced and the final construct digested with PacI enzyme to target homologous recombination to the correct chromosomal safe-haven locus.

### Mad1 alleles (T661A, T667A, T668A, TT667,668AA and RLK-AAA)

The HISp-GFP-Mad1 plasmid was mutagenized using the Quickchange lightning kit (Agilent), according to the manufacturer's instructions.

### GAL7p-myc-MAD2

*MAD2* was ectopically expressed to rescue the *mad2Δ* phenotype. Ectopic expression of myc-tagged Mad2 was generated under the $P_{GAL7}$ promoter. The *GAL7* promoter, myc tag (EQKLI-SEEDLN) and *MAD2* ORF with its 3'UTR were assembled into pPEE37 (HYG resistance, chromosome 3 safe-haven) using Gibson assembly.

### MAD2p-MAD2

For Fig 3B a genomic *MAD2* clone, with its own promoter, was used to rescue the *mad2Δ*.

### Cse4-mCherry

Cse4 was tagged with mCherry at the N-terminus and ectopically expressed for use as a centromere marker. This construct was generated under the endogenous Cse4 (1kb) promoter. The promoter and full length *CSE4* clone were PCR-amplified from H99 genomic DNA. These two fragments along with amplified mCherry were cloned into vector pPEE31 (NAT resistance, chromosome 14, safe-haven 7 near CNAG 05557) using Gibson assembly.

### lacO array

A 240xlacO array was released from pLAU43 [33] as an Nhe1-Xba1 fragment, and then blunt end ligated into the pPEE37 vector. After linearisation this DNA was transformed into the wild-type H99 strain, to integrate at the safe-haven on chromosome 3.

### HISp-mNeonGreen-lacI

A *Cryptococcus*-optimised mNeonGreen ORF, containing an intron to boost expression, was synthesised. Codon usage for mNeonGreen was randomly generated from the codon frequency table for the most translated 5% of CDS in H99 (PMID: 32020195). Intron 1 from CNAG_05249, with a modified 5' (GTATGT) and 3' (CAG) splice site designed by Huang et al. (PMID: 34791226), was inserted into mNeongreen after Lys50. This was then combined with the HIS promoter, lacI ORF and pPEE36 vector (G418 resistance, chromosome 14, targeting safe-haven 7 next to CNAG_05557), through Gibson assembly.

### GALp-Myc-MPS1

Mps1 was over-expressed ectopically using a *GALp-myc-MPS1* construct. Endogenous *MPS1* and Gal7 promoter were PCR-amplified from H99 genomic DNA. All three fragments were cloned into vector pPEE36 (G418 resistance, chromosome 14, targeting safe-haven 7 next to CNAG_05557) using Gibson assembly.

### Microscopy

Live-cell microscopy was performed with a spinning disc confocal microscope (Nikon Ti2 CSU-W1) with a 100X oil objective (Plan Apo VC) coupled to a Teledyne-Photometrics 95B sCMOS camera. For imaging, Z-stacks of 11 images (step size 0.5μm) were acquired using a 491nm laser line for GFP and 561nm laser for mCherry. Exposure times were typically 300ms

and laser power was kept to the minimum to avoid photobleaching. Images were captured using Nikon Elements software.

ImageJ was used for image analysis, which were then further processed in Adobe Photoshop to adjust brightness and contrast. All adjustments were applied to whole images uniformly, and to all images being compared.

## Checkpoint assays

**Benomyl plates—serial dilution assay.**   Cells from an overnight culture were diluted to $OD_{600}$ ~0.4 in distilled water. Then 10-fold, serial dilutions were made and spotted onto YPDA plates (with or without the anti-microtubule drug, benomyl at different concentrations) then typically incubated at 30˚C for 48 hours.

Benomyl stock was 30mg/ml in DMSO and, due to solubility issues, this was added directly to boiling YPD agar.

**Re-budding assays.**   Glycerol stocks of *Cryptococcus* strains were stored at -80˚C. Cells were streaked out on YPDA plates and allowed to grow on plates for two days at 30˚C. After two days, cells were grown overnight (to $OD_{600}$ of ~0.5) in 500mls of YPDA media. For mitotic arrest, 2.5µg/ml nocodazole was added to the cells and incubated for three hours. Cells were harvested by centrifugation at 5000 rpm at room temperature, for 3 mins. Following this, cells were washed with distilled water twice and mounted on slides for microscopy. Careful microscopic observation has been made to determine percentage of mitotic arrested cells with large buds. Cells with bud size greater than 4µm has been categorized as 'Large budded' mitotic arrested cells. While cells having daughters size ranging from 0.5–4µm were categorized as 'small budded'. From microscopic images of fixed cells, the percentage of large-budded cells were calculated and compared between wild type and knockout strains. This experiment was repeated three times with similar results. 300 cells were counted per strain, in each experiment. GraphPad prism version 8 was used for statistical analyses.

**Re-budding assay in microfluidics.**   We used Alcatras microfluidic cell traps incorporated into a device allowing for use with five strains [55]. We moulded devices in polydimethylsiloxane (PDMS) from an SU8-patterned wafer with an increased thickness of 7µm, to accommodate the larger size of *C neoformans* cells compared to *S cerevisiae* (manufactured by Microresist, Berlin, design available on request). Imaging chambers for individual strains are isolated by arrays of PDMS pillars separated by 2µm gaps. This prevents intermixing of strains while cells experience identical media conditions.

Before use we filled the devices with synthetic complete (SC) media, supplemented with 0.2g/l glucose and containing 0.05%w/v bovine serum albumin (Sigma) to reduce cell-cell and cell-PDMS adhesion. Cells pre-grown to logarithmic phase in the same media (lacking the BSA) were injected into the device. An EZ flow system (Fluigent) delivered media at 10µl per minute to the flow chambers and performed the switch to media containing nocodazole after 5 hours. This media also contained Cy5 dye to allow monitoring of the timing of the media switch. We captured image stacks at 5-minutes intervals at 4 stage positions for each strain, using a Nikon TiE epifluorescence microscope with a 60x oil-immersion objective (NA 1.4), a Prime95b sCMOS camera (Teledyne Photometrics) and OptoLED illumination (Cairn Research). Image stacks had 5 Z-sections, separated by 0.6µm, captured using a piezo lens positioning motor (Pi).

**Sister-chromatid separation assay.**   We observed the dynamics of chromosome three of *Cryptococcus* strains in living cells using *lacO*-mNeonGreen-lacI system. This system has been adapted for use in *Cryptococcus neoformans* by expressing mNeonGreen-lacI under the Gpd1 promoter. We targeted integration of the *lacO* array in chromosome three at safe-haven 3 near

CNAG_02589. Cells were grown to $OD_{600}$ of ~0.4 in 500mls of YPDA. For mitotic arrest, 2.5μg/ml nocodazole was added to the cells and incubated for three hours. Cells were harvested by centrifugation at 5000 rpm at room temperature, for 3 mins. Following this, cells were washed with distilled water twice and mounted on slides for microscopy. Careful microscopic observation determined if separation of two mNeonGreen-lacI dots representing the replicated sister chromatids had occurred or not.

**Mps1 overexpression assays.** *Cryptococcus* strains were grown overnight (to $OD_{600}$ of ~0.5) in 50 mls of YP media. Next morning, 2% galactose was added to the cultures and incubated for a further three hours. Cells were then harvested by centrifugation at 5000 rpm at room temperature, for 3 mins. Following this, cells were washed with distilled water twice and mounted on slides for microscopy. From microscopic images, metaphase arrested cells with short mitotic spindles (Fig 6A) or single, DAPI-stained nuclei in the bud (Fig 6D) were counted and analysed using GraphPad prism.

**Immunoblot analysis.** For whole-cell lysates, typically a 10 ml ($OD_{600}$ ~0.5) cell culture was harvested by centrifigation. Cells were washed once and snap frozen in liquid nitrogen. Frozen pellets were resuspended in 2x sample buffer with 200mM DTT and 1mM PMSF. Cells were then disrupted with 0.5mm Zirconia/Silica beads (Thistle Scientific) using a multi-bead beater for 1 min (BioSpec Products). Samples were spun, boiled for 5 min at 95˚C, and the cell debris pelleted by centrifugation for 5 min at 13000 rpm. Cleared extracts were then immediately loaded and separated by SDS-PAGE. Proteins were transferred onto nitrocellulose membrane (Amersham Protran 0.2 μm nitrocellulose, GE Healthcare Lifescience) using a semi-dry transfer unit (TE77, Hoefer, Inc, MA, USA) in 25mM Tris, 130 mM glycine and 20% methanol. Transfer was typically for 1.5–2 hours at 150–220 mA. Efficiency of protein transfer was visualised using Ponceau S solution. Membranes were blocked with Blotto (PBS, 0.04% Tween 20, 4% Marvel skimmed milk powder) for at least 30 min at room temperature on a shaking platform. The primary antibody (anti-Mad1, anti-GFP or anti-Myc [9E10 or 9B11, Cell Signalling]) was incubated in the same blocking buffer (1 in 1000 dilution) overnight at 4˚C. The membrane was then washed 3 times for 10 mins with PBS+0.04% Tween and then incubated with the HRP-coupled secondary antibody (1 in 5000 dilution) for at least an hour at room temperature. The membrane was washed again, rinsed with PBS and ECL performed (SuperSignal West Pico, or SuperSignal West Femto, Thermo Fisher Scientific Inc, IL, USA).

**Mps1 purifications and kinase assays.** Residues 478–784 of CnMps1 were amplified from cDNA and cloned into the 14S Biobrick vector. Induction of protein expression was performed in BL21 (pLysS) cells. IPTG was added and cultures incubated for 16 hrs at 18˚C. Cells were harvested, washed and pellets frozen in liquid nitrogen. Cell pellets were resuspended in lysis buffer [50mM Tris-HCl pH8, 500mM NaCl, 10% glycerol, 5mM imidazole, 1mM β-mercaptoethanol, EDTA-free protease inhibitor tablet (Roche), 1mM PMSF] then lysed by sonication (1 sec ON and 2sec OFF for a total of 3 min). To remove the cell debris, lysed cells were centrifuged at 20,000 rpm, for 30–45 min, at 4˚C, and the lysate filtered through a 0.45μm syringe. Lysates were then incubated with rotation for 2 hours (at 4˚C) with Talon cobalt resin (Thermofisher). After incubation, the beads were transferred to a Biorad column, washed with 10 column volumes of wash buffer, and protein eluted with lysis buffer containing 250mM imidazole. The recombinant kinase domain was dialysed overnight into 50mM Tris-HCl pH8, 150mM NaCl, 5% glycerol, 2mM DTT and proteins concentrated via centrifugation (Vivaspin, Sartorius). Recombinant kinase was added to 10μl of 2X kinase buffer [40mM Hepes (pH 7.5), 200mM KCl, 20mM $MgCl_2$, 2mM DTT, 400μM ATP], substrate, and water to a final volume of 20μl. Reactions were incubated at 30˚C for 30 min and quenched with an equal volume of SDS-PAGE sample buffer and resolved by SDS-PAGE.

Mad1-CTD substrate: residues 324–679 of Mad1 were amplified from cDNA and cloned into a LIC Biobrick vector (14C, 6xHis-MBP). This Mad1 construct was expressed in *E. coli* pLysS cells (Agilent), purified on Talon Cobalt resin, eluted and then dialysed into 50mM Hepes pH7.6, 75mM KCl.

Radioactive Mps1 kinase assays were performed for 30 min at 30˚C: 20mM Hepes (pH 7.5), 100mM KCl, 10mM $MgCl_2$, 1mM DTT, 100µM cold ATP, 5µCi γ-$^{32}$P-labelled-ATP and 1µg substrate.

**Lysis of large-scale cell extracts for mass spectrometry.**   Yeast cells were grown overnight (to $OD_{600}$ of ~0.5) in 500mls of YPDA. 2.5µg/ml nocodazole was added to the cells and incubated for three hours. Cells were harvested by centrifugation at 5000 rpm at 4˚C, for 15 mins. Pelleted cells were frozen in drops, using liquid nitrogen. The cells were then ground manually, using a ball grinder. Yeast powders were resuspended into lysis buffer containing 50mM Hepes pH7.6, 75mM KCl, 1mM MgCl2, 1mM EGTA, 10% Glycerol, 0.1% Triton X-100, 1mM $Na_3VO_4$, 10 µg/mL CLAAPE (protease inhibitor mix containing chymostatin, leupeptin, aprotinin, antipain, pepstatin, E-64 all dissolved in DMSO at a final concentration of 10 mg/mL), 1 mM PMSF, 0.01 mM microcystin. 1g of yeast powder was resuspended in 1ml of the lysis buffer. Cell lysis was completed by sonication (cycles of 5 sec ON, 5 sec OFF for 1 min). After sonication, the cell debris was pelleted (30 min, at 22000 rpm, at 4˚C) and the supernatant incubated with anti-GFP TRAP magnetic agarose beads (ChromoTek) for 1 hr at 4˚C. The beads were washed at least 9 times with wash buffer (50mM Hepes pH7.6, 75 mM KCl, 1 mM MgCl2, 1 mM EGTA, 10% Glycerol) and once with PBS+0.001% Tween 20. Proteins were eluted from the beads by adding 2X sample buffer containing 200mM DTT and boiled at 95˚C for 5–10 min, before running on an SDS-PAGE gel.

**GFP-Mad1 mass-spectrometry and volcano plots.**   Protein samples from all biological replicates were processed at the same time and using the same digestion protocol without any deviations. They were subjected for MS analysis under the same conditions, and protein and peptide lists were generated using the same software and the same parameters. Specifically, proteins were separated on gel (NuPAGE Novex 4–12% Bis-Tris gel, Life Technologies, UK), in NuPAGE buffer (MES) and visualised using Instant Blue stain (AbCam, UK). The stained gel bands were excised and de-stained with 50mM ammonium bicarbonate (Sigma Aldrich, UK) and 100% (v/v) acetonitrile (Sigma Aldrich, UK) and proteins were digested with trypsin, as previously described 69. In brief, proteins were reduced in 10mM dithiothreitol (Sigma Aldrich, UK) for 30mins at 37ÅãC and alkylated in 55mM iodoacetamide (Sigma Aldrich, UK) for 20 min at ambient temperature in the dark. They were then digested overnight at 37ÅãC with 12.5 ng trypsin per µL (Pierce, UK). Following digestion, samples were diluted with an equal volume of 0.1% TFA and spun onto StageTips as described previously 70. Peptides were eluted in 40 µL of 80% acetonitrile in 0.1% TFA and concentrated down to 1 µL by vacuum centrifugation (Concentrator 5301, Eppendorf, UK). The peptide sample was then prepared for LC-MS/MS analysis by diluting it to 5 µL with 0.1% TFA.

LC-MS analyses were performed on an Orbitrap Exploris 480 Mass Spectrometer (Thermo Fisher Scientific, UK) coupled on-line to an Ultimate 3000 HPLC (Dionex, Thermo Fisher Scientific, UK). Peptides were separated on a 50 cm (2 µm particle size) EASY-Spray column (Thermo Scientific, UK), which was assembled on an EASYSpray source (Thermo Scientific, UK) and operated constantly at 50oC. Mobile phase A consisted of 0.1% formic acid in LC-MS grade water and mobile phase B consisted of 80% acetonitrile and 0.1% formic acid. Peptides were loaded onto the column at a flow rate of 0.3 µL min-1 and eluted at a flow rate of 0.25 µL min-1 according to the following gradient: 2 to 40% mobile phase B in 150 min and then to 95% in 11 min. Mobile phase B was retained at 95% for 5 min and returned to 2% a minute after until the end of the run (190 min). Survey scans were recorded at 120,000 resolution

(scan range 350–1500 m/z) with an ion target of 4.0e5, and injection time of 50ms. MS2 was performed in the orbitrap at 15,000 resolution, with ion target of 2.0E4 and HCD fragmentation (Olsen et al, 2007) with normalized collision energy of 28. The isolation window in the quadrupole was 1.4 Thomson. Only ions with charge between 2 and 6 were selected for MS2. Dynamic exclusion was set at 60 s.

The MaxQuant software platform 71 version 1.6.1.0 was used to process the raw files and search was conducted against our in-house *Cryptococcus neoformans var. grubii* protein 25 database, using the Andromeda search engine 72. For the first search, peptide tolerance was set to 20 ppm while for the main search peptide tolerance was set to 4.5 pm. Isotope mass tolerance was 2 ppm and maximum charge to 7. Digestion mode was set to specific with trypsin allowing maximum of two missed cleavages. Carbamidomethylation of cysteine was set as fixed modification. Oxidation of methionine, and phosphorylation of serine, threonine and tyrosine were set as variable modifications. Label-free quantitation analysis was performed by employing the MaxLFQ algorithm 73. Peptide and protein identifications were filtered to 1% FDR. Statistical analysis was performed by Perseus software 74, version 1.6.2.1.

Volcano plots show both the differences (mean, label free quantitation LFQ difference, on the x-axis) and their statistical confidence ($-\log_{10}$P-value of Perseus statistical test, on the y-axis) between polypeptides identified in two datasets (n = 3 for each purification). The $-\log_{10}$P-value of 1.3 used here corresponds to a p-value cutoff of 0.05.

**Computational structural analysis.** A three-dimensional structural model for *Cryptococcus neoformans* Mad1 was obtained using a ColabFold version that employs AlphaFold2 with MMseqs2 [56]. AlphaFold2 generated *Cryptococcus neoformans* Mad1 structure was compared with human Mad1 C-terminal domain (PDB: 4DZO) using PyMOL (The PyMOL Molecular Graphics System, Schrödinger, LLC).

## Supporting information

**S1 Fig. The *MAD1* locus was correctly targeted. (A) Targeting schematic. (B)** PCR analysis of genomic DNA confirms that the Blaster cassette had integrated at the *CnMAD1* locus.
(TIF)

**S2 Fig. The MAD2 locus was correctly targeted. (A) Targeting schematic. (B) PCR analysis of genomic DNA. This confirms that the Blaster cassette had integrated at the CnMAD2 locus, and that the amdS marker recombined out appropriately.**
(TIF)

**S3 Fig. *mad* and *mps1* mutants die faster than wild-type cells in nocodazole.** The four strains were grown to log phase and then nocodazole added to a final concentration of 2μg/ml. Cultures were then washed, diluted and plated onto YPD. Viable colonies were counted after 3 days growth.
(TIF)

**S4 Fig. GFP-Mad1 localises to kinetochores early in mitosis.** In nocodazole arrested cells, GFP-Mad1 does co-localise with the kinetochore marker mCherry-Dad2. In cycling cells they only co-localise early in mitosis (cell marked with arrow head), not in late mitosis (cell marked with *). Scale bar is 10μm.
(TIF)

**S5 Fig. Mad1 pulls down the TPR nucleoporin in interphase.** Volcano plots show the difference (mean LFQ difference) and confidence ($-\log_{10}$P-value of Perseus statistical test) between the cycling and nocodazole-arrested GFP-Mad1 pull-downs (n = 3 for each). This is the same

dataset as in Fig 5B. Here nucleoporins are highlighted, some of which including TPR are enriched with the Mad1 pulldown in non-mitotic cells.
(TIF)

**S6 Fig. *mad1-567RLK/3A* and *549RLK/3A* phenotypes.** The conserved RLK motif in Mad1 (residues 567–9) is required for Bub1 binding (Fig 5E) and when mutated generates a benomyl sensitive strain. Mad1 RLK residues 549–551 are not conserved, are not required for Bub1 binding (Fig 5E) and do not produce a benomyl-sensitive strain when mutated. The strains indicated were serially diluted and plated onto YPD plates with and without benomyl. Images were taken after 3 days growth at 30.
(TIF)

**S7 Fig. (A) *in vitro* Mad1 phosphorylation sites detected by mass spectrometry.** The C-terminus of Mad1 was phosphorylated *in vitro* by Mps1 kinase, run on a gel and the Mad1 band excised then digested with trypsin. The phosphopeptides identified are listed and highlighted in yellow on the sequence of Mad1p. The probabilities of specific S/T residue modification are indicated. **(B) The *mad1* phosphomutant proteins are stable.** Immunoblot of whole cell extracts from GFP-mad1 phosphomutants, using the anti-Mad1 antibody. * indicates a cross-reacting band, used here as a loading control.
(TIF)

## Acknowledgments

We would like to thank all members of the Hardwick and JP labs for their support, discussions and suggestions on this manuscript; Liz Ballou for many helpful *Cryptococcus* tips and suggestions; Paige Erpf and James Fraser for Safe Haven and Blaster constructs; Kaustuv Sanyal, Lukasz Kozubowski, Yong-Sun Bahn and Liz Ballou for *Cryptococcus neoformans* strains and plasmids; David Leach and Dave Sherratt for the lacO array; Ken Sawin for anti-mCherry antibodies; Connie Nichols for information on pCN19; Dave Kelly, Toni McHugh and Dhanya Cheerambathur for help with microscopy and generating scripts, and Peter Swain for leading on the ISSF funding (IC).

## Author Contributions

**Conceptualization:** Koly Aktar, Kevin G. Hardwick.

**Data curation:** Christos Spanos.

**Formal analysis:** Koly Aktar, Ioanna Leontiou, Kevin G. Hardwick.

**Funding acquisition:** A. Arockia Jeyaprakash, Kevin G. Hardwick.

**Investigation:** Koly Aktar, Thomas Davies, Ioanna Leontiou, Ivan Clark, Christos Spanos, Kevin G. Hardwick.

**Methodology:** Koly Aktar, Thomas Davies, Ioanna Leontiou, Ivan Clark, Edward Wallace, A. Arockia Jeyaprakash, Kevin G. Hardwick.

**Project administration:** Kevin G. Hardwick.

**Resources:** Edward Wallace, Laura Tuck, A. Arockia Jeyaprakash.

**Software:** Ivan Clark.

**Supervision:** Ioanna Leontiou, Kevin G. Hardwick.

**Visualization:** Thomas Davies, Ivan Clark, Christos Spanos, A. Arockia Jeyaprakash, Kevin G. Hardwick.

**Writing – original draft:** Koly Aktar, Kevin G. Hardwick.

**Writing – review & editing:** Thomas Davies, A. Arockia Jeyaprakash, Kevin G. Hardwick.

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
