## [Decision Letter · Decision Letter 0]

27 Sep 2023

Dear Dr Hardwick,

Thank you very much for submitting your Research Article entitled 'mps1 and mad mutations reduce Cryptococcus neoformans titan cell viability' to PLOS Genetics.

The manuscript was fully evaluated at the editorial level and by independent peer reviewers. The reviewers appreciated the attention to an important problem, but raised some substantial concerns about the current manuscript. Based on the reviews, we will not be able to accept this version of the manuscript, but we would be willing to review a much-revised version. We cannot, of course, promise publication at that time.

If you decide to revise the manuscript for further consideration at PLOS Genetics, please aim to resubmit within the next 60 days, unless it will take extra time to address the concerns of the reviewers, in which case we would appreciate an expected resubmission date by email to plosgenetics@plos.org.

We are sorry that we cannot be more positive about your manuscript at this stage. Please do not hesitate to contact us if you have any concerns or questions.

Yours sincerely,

Anna Selmecki, Ph.D.

Academic Editor

PLOS Genetics

Geraldine Butler

Section Editor

PLOS Genetics

Reviewer's Responses to Questions

**Comments to the Authors:**

Reviewer #1: In the manuscript by Aktar K. et al., the authors characterize the Spindle Assembly Checkpoint (SAC) in Cryptococcus neoformans. The Authors focus on the Mps1 kinase and the downstream checkpoint components Mad1 and Mad2. The Authors provide multiple lines of evidence using complementary experiments that these components act as part of SAC in this species. The Authors conclude that the classic checkpoint functions of the SAC are conserved in Cryptococcus. The Authors also perform purification of GFP-Mad1 followed by mass spectrometric analysis of associated proteins, which reveals not only expected SAC components but also other proteins that can be further investigated. Finally, the Authors quantitated viability of titan cells and their daughters and find that the mad1, mad2 and mps1 mutants exhibit reduced tian cell/titan daughter viability. The Authors propose that C. neoformans Mps1, Mad1 and Mad2 proteins have important, possibly novel, roles in ensuring high fidelity chromosome segregation during stressful conditions.

Comments

1. In the abstract the Authors state that cells lacking Mad1 or Mad2 when treated with microtubule depolymerizing agent “die rapidly”. I suggest the Authors use a more precise way to describe the timing of death, as the time of death is relative.

2. The Authors state: “Fig.1D shows that we can rescue the benomyl sensitivity of the mad1� strain by complementation with a GFP-Mad1 construct”. I suggest the Authors also add here that this also means that the GFP-Mad1 chimera is likely at least partially functional (that GFP did not abrogate the function of Mad1 when used as a fluorescent tag on Mad1).

3. In Fig 2D, mad1D on 2.5 concentration of Ben shows some spontaneous mutants that show colony growth – I am curious to see what are those? Any precedence for this from other yeasts?

4. Why did the Authors switched between benomyl and nocodazole in some experiments? It should be explained what is the benefit of using one depolymerizing drug versus the other and not thiabendazole for instance.

5. In Figure 3B graph, I suggest, the Authors label the Y axis as % large-budded cells and not mention it is an arrest - especially for the mutants it is not known whether those observed large-budded cells are indeed arrested or simply at metaphase-anaphase and continuing the cell cycle.

6. When referring to Fig.S2, the Authors state: “The mad and mps1 mutants all display a rapid rate of death” I suggest the Authors use more precise description instead of “rapid”.

7. I suggest for the figure S2 graph, in the label of the X axis, the Authors include "Length of exposure to nocodazole (minutes)”

8. The Authors state: “nuclear pores act as a site of MCC”. I suggest the Authors define the MCC here (Mitotic checkpoint complex).

9. Figure 5 C and D (especially D) need DIC/brightfield images so that individual cells can be identified.

10. Is the nuclear periphery localization of Mad1 the same as previously described localization of CENP-A in pre-mitotic C. neoformans cells (does Mad1 localize to centromeres in pre-mitotic/stationary phase cells?)? Even though the Authors test co-localization of Mad1 and Cse4, they don't show non-arrested stationary cells and don't comment on this question.

11. The Authors state: “We grew 500mls of yeast in YPD” – space is missing between 500 and mls – mls should be ml?

12. The Authors state: “We were surprised to observe that all of these strains (wild-type, mad1 and mad2 died quickly after galactose addition (data not shown).” These two findings are intriguing but not explored in this manuscript. This observation also correlates with and may be related to the low viability of WT cells treated with nocodazole as shown in FigS2.

13. The Authors state: “We conclude that the C-terminal phosphorylation site T667 is an important Mps1 substrate in C. neoformans.” – Up to this point in the manuscript, all the data are robust and well presented but not surprising or novel - C. neoformans SAC seems to function in principle very similar to other already described systems including other (ascomycetous) yeasts. The element of novelty is the fact that this work presents for the first time SAC in basidiomucetous fungus.

14. The Authors state: “The same experiment reveals that only ~50% of mps1 mutant titan cells were viable and around ~70% of the mad1 and mad2 mutant titans.” I suggest adding … “were viable” at the end of the sentence or moving “were viable” to the end of the sentence. Also, 70% viability of mad1 and mad2 mutants should not be classified as a severe defect. While the defect in mps1 delta cells is more dramatic, defect observed for mad mutants is not severe.

15. The Authors state: “Our interpretation of this experiment is that at least a sub-set of the checkpoint proteins (Mad1, Mad2 and particularly Mps1 kinase) have a novel function that is particularly important for the viability of titan cells and/or their daughters.” This is very intriguing but is not followed upon in this study - it remains unclear what is this presumed function.

16. Regarding temperature sensitivity testing of the mad and mps1 mutants - Perhaps testing in on plates at 39C may show more striking phenotype. Also, it would be interesting to test what happens to those mutant cells upon exposure to 37C/39C? Do they die, or are just arrested?

17. Regarding the experiment and data presented in Fig.9 - As these mutant cells are generally growing poorly at 37C (and potentially may have even more severe growth defect when 5% CO2 and poor nutrient environment is applied), it would be important to ask if upon exposing the cells to titanization protocol also the non-titans are less viable/dead. The Authors micromanipulated and tested specifically the titans but did not test regular cells. It is possible that the observed loss of viability is common to all cells and not specifically titans.

18. The Authors state: “Point mutations that abolish the checkpoint, such as mad1-T667A and bub1-cd1, did not display reduced titan cell viability.” Are those specific two mutants temperature sensitive?

19. The Authors state: “The importance of MAD1, MAD2 and MPS1 in titan cells could be directly related to their increased ploidy.” This is an intriguing idea - however, it should be first tested if non-titans of the SAC defective mutants have reduced viability after the titanization assay (as described in #17) and whether there is significant difference between non-titans and titans in this assay.

20. In some parts of text, space is lacking between the number and the unit - for instance in Figure 2C and D panels.

Reviewer #2: This paper deals with the characterization of some spindle checkpoint proteins (mainly Mad1) in the pathogenic yeast Cryptococcus neoformans. The authors identified MAD1 and MAD2 on the basis of sequence homology and carried out their first functional characterization. As previously shown in other systems, C.n. Mad1 and Mad2 are required for the metaphase arrest in the presence of microtubule-depolymerizing drugs or upon MPS1 overexpression. Additionally, Mad1 localizes at nuclear pores in interphase and relocalizes to kinetochores in mitosis. The study of C.n. Mad1 interactors by mass spectrometry identified several proteins that are known to interact with Mad1 in other systems (e.g. Bub1, Mad2, Cdc20, APC subunits, etc.). Authors identified an RLK motif that is required for Mad1 interaction with Bub1 and for normal benomyl sensitivity (suggesting a likely involvement in the checkpoint). They also identified Mps1-dependent phosphorylation sites in vitro, one which may be important to induce a metaphase arrest upon nocodazole treatment or MPS1 overexpression. Finally, they show that deletion of MAD1, MAD2 or MPS1 significantly impacts the viability of titan cells, which are specialized polyploid cells that C.n. cells produces during human infection and are linked to pathogenicity and drug resistance. This raises the possibility that spindle checkpoint genes may be exploited as potential therapeutic targets.

The paper is well written and the logic easy to follow. Experiments are generally well designed and carefully interpreted. There are however a few points that in my opinion should be addressed to consolidate some observations and strengthen the conclusions.

Main points:

The data about Mad1 phosphorylation by Mps1 are interesting, but experiments supporting that phosphorylation takes place also in vivo are missing. Since the authors have raised anti-Mad1 antibodies, they could test if Mad1 undergoes a phosphorylation-dependent shift in electrophoretic mobility upon nocodazole treatment, as previously shown by the senior author for budding yeast Mad1. Alternatively, the mass spec data on GFP-Mad1 IPs could be further analyzed to map Mad1 phosphorylation sites and compare them with those obtained in vitro.

In Fig. 6A the graph label (cycling vs noc arrested GFP-Mad1) does not match the text on page 8 (tagged vs untagged). If the data refer to the cycling vs noc-arrested condition, which I believe is the case, what is the difference with Fig. 6C? The two graphs look exactly the same.

Also, was the untagged control arrested in nocodazole? The way cells were grown and treated should appear in the figure legend.

Western blots (Fig. 7C, 7E, 8E) lack a loading control. For instance, in Fig. 7E it seems that MAD1 deletion appreciably reduces the levels of MPS1 overexpression. Is this the case?

Minor points:

In Fig. 1D labeling of the first image should be “no benomyl” or “0 ug/ml” instead of “30°C”. Indeed I assume that all plates were incubated at 30°C. Same comment for Fig. 2D, 8D and S3.

The style of the two graphs in Fig. 2E should be homogenized.

I assume that Fig. 5B shows the same cell at different stages of mitosis. Wouldn’t it be more logical to invert the order of the images, so that mitosis proceeds from top to bottom, rather than the other way around? Why does GFP-Mad1 show a diffuse signal rather than being at nuclear pores in the small budded cell at the bottom raw?

Were the IPs in Fig. 6F done from nocodazole-treated cells? This information should appear in the figure legend.

The Y axis in Fig. 7D is unlabeled.

In Fig. 8A it is not clear which band corresponds to His-Sumo-Mps1. What is the purpose of having a Sumo tag?

Was the graph in Fig. 8B normalized to the amount of Mad1 in each sample (note that Mad1-CT-AA has lower levels than Mad1-CT)?

In Fig. 8F the data from the three biological replicates are more dispersed for the T667A mutant than for T667-8A, yet the latter has four asterisks of statistical significance while the latter has only three. Is this correct?

Fig. 9C lacks error bars.

In Fig. S2 mutants are defined as KO, as opposed to � in the text and figure legend.

Reviewer #3: In a very nice study, the authors investigate the mechanisms of the spindle assembly checkpoint (SAC) in the fungal pathogen Cryptococcus neoformans. The authors first identify genes in C. neoformans that are orthologous to the SAC effectors MAD1, MAD2, and MPS1. Using microtubule poisons, they go on to demonstrate the requirements for MAD1, MAD2, and MPS1 to arrest cells in response to spindle perturbations. Using co-localization studies and genetic perturbation of specific phosphorylation sites, the authors develop evidence for mechanistic interactions

---

## [Decision Letter · Decision Letter 1]

28 Mar 2024

Dear Dr Hardwick,

Thank you very much for submitting your Research Article entitled 'Conserved signalling functions for Mps1, Mad1 and Mad2 in the Cryptococcus neoformans spindle checkpoint' to PLOS Genetics.

The manuscript was fully evaluated at the editorial level and by independent peer reviewers. The reviewers appreciated the attention to an important topic but identified some concerns that we ask you address in a revised manuscript.

We therefore ask you to modify the manuscript according to the review recommendations. Your revisions should address the specific points made by each reviewer.

Yours sincerely,

Anna Selmecki, Ph.D.

Academic Editor

PLOS Genetics

Geraldine Butler

Section Editor

PLOS Genetics

Dear Dr. Hardwick,

Thank you for your patience as we discussed the reviewers' comments. There are several minor changes that need to be addressed, given the removal of the Titan cells and focus on the SAC. If you wish to provide a revised manuscript the Editors will review the updated manuscript.

Reviewer's Responses to Questions

**Comments to the Authors:**

Reviewer #1: The Authors, Aktar K. et al. have addressed all the points/comments I’ve included in my previous review. The revised manuscript is further improved and overall, this is a thorough study that is well put together. The Authors decided to remove the data that relate to the question of the impact of the SAC on formation of titan cells. This leaves the manuscript mostly focused on characterizing the SAC major architecture. The main message in the revised version is that the SAC architecture is conserved in C. neoformans. Below are some additional comments.

1. In the Abstract there are double “that that”

2. A comment to the Authors Summary statement: a pathway that is "extremely well conserved" may not be a good choice for antifungal therapy.

3. A considerably large fraction of the Introduction is devoted to Titan cells - as the revised manuscript does not address titan cells, I suggest this fragment is reduced and other aspects are included, for instance the fact that C. neoformans acquires resistance to fluconazole via becoming an aneuploid (which is not necessarily related to formation of titan cells) and SAC may be involved in this process.

4. In the Introduction: I am not sure if emphasizing SAC as a potential drug target is justified here, especially given its apparently high level of conservation.

5. Fig 1B shows that the GFP-Mad1 is functional while Fig 1C shows that the expression levels of GFP-Mad1 are similar to wt Mad1. To better judge the latter, a loading control would be helpful.

6. Why did the complementation of mad2 mutant growth require expression of MAD2 from GAL promoter?

7. In Fig 1, the growth of the double mutant is assessed in panel F, not E.

8. A mild ts phenotype of the mad1 and mad2 mutants may be more pronounced at 39C

9. In Fig S3, it is striking that the WT is much affected by the nocodazole arrest. Was the WT and the mutants also tested with exposure to benomyl instead of nocodazole? The mutants exposed to nocodazole only for 25 min already loose viability dramatically. Is this expected? Or alternatively, there is some additional effect of nocodazole that lowers cell viability. Those comments do not change/question the main conclusion but rather point to an additional phenomenon that may be of interest.

Reviewer #2: The authors addressed satisfactorily most of my comments. I just have a couple of minor comments on the text that the authors may want to address before final acceptance.

In Fig. 5E, were the immunoprecipitates split in two different gels to be probed with anti-GFP and anti-Mad1 antibodies? If this is the case it should be specified in the figure legend.

In Fig. 6E, is the anti-myc Ab really 16B12 or rather 16B11? I think that 16B12 is an anti-HA Ab.

In Fig. S4 it is not clear why Dad2-mCherry is not visible in all cells. A possible explanation could be given in the figure legend. The addition of arrowheads in the images would help the readers to identify early mitotic and anaphase cells.

Fig. S6 seems to have two titles.

Reviewer #3: The authors provided a very thoughtful and comprehensive response to reviewer comments, and I am quite satisfied with the updates. I have no additional critiques of the manuscript.

**Have all data underlying the figures and results presented in the manuscript been provided?**

Reviewer #1: Yes

Reviewer #2: Yes

Reviewer #3: Yes

PLOS authors have the option to publish the peer review history of their article (what does this mean?). If published, this will include your full peer review and any attached files.

Reviewer #1: No

Reviewer #2: **Yes: **Simonetta Piatti

Reviewer #3: **Yes: **Steven B Haase

---

## [Editor Report · Decision Letter 2]

14 May 2024

Dear Dr Hardwick,

We are pleased to inform you that your manuscript entitled "Conserved signalling functions for Mps1, Mad1 and Mad2 in the Cryptococcus neoformans spindle checkpoint" has been editorially accepted for publication in PLOS Genetics. Congratulations!

Yours sincerely,

Anna Selmecki, Ph.D.

Academic Editor

PLOS Genetics

Geraldine Butler

Section Editor

PLOS Genetics

Comments from the reviewers (if applicable):

**Data Deposition**

http://datadryad.org/submit?journalID=pgenetics&manu=PGENETICS-D-23-00900R2

**Press Queries**

---

## [Editor Report · Acceptance letter]

28 May 2024

PGENETICS-D-23-00900R2 

Conserved signalling functions for Mps1, Mad1 and Mad2 in the Cryptococcus neoformans spindle checkpoint 

Dear Dr Hardwick, 

We are pleased to inform you that your manuscript entitled "Conserved signalling functions for Mps1, Mad1 and Mad2 in the Cryptococcus neoformans spindle checkpoint" has been formally accepted for publication in PLOS Genetics! Your manuscript is now with our production department and you will be notified of the publication date in due course.

With kind regards,

Olena Szabo

PLOS Genetics

On behalf of:
